# The invisible plant technology of Prehistoric Southeast Asia: Indirect evidence for basket and rope making at Tabon Cave, Philippines, 39–33,000 years ago

Hermine Xhauflair[1,2]*, Sheldon Jago-on[3●], Timothy James Vitales[3●], Dante Manipon[1●], Noel Amano[4●], John Rey Callado[3●], Danilo Tandang[3●], Céline Kerfant[5●], Omar Choa[2], Alfred Pawlik[6]

1 School of Archaeology, University of the Philippines Diliman, Quezon City, Philippines, 2 UMR 7194 of the CNRS, Département Homme et Environnement, Muséum National d'Histoire Naturelle and Université de Perpignan Via Domitia, Institut de Paléontologie Humaine, Paris, France, 3 National Museum of the Philippines, City of Manila, Philippines, 4 Department of Archaeology, Max Planck Institute of Geoanthropology, Jena, Germany, 5 Departament d'Humanitats, Història / Culture and Socio-Ecological Dynamics (CaSEs), Universitat Pompeu Fabra, Barcelona, Spain, 6 Department of Sociology and Anthropology, School of Social Sciences, Ateneo de Manila University, Quezon City, Philippines

● These authors contributed equally to this work.
* hermine_xhauflair@hotmail.com

**Data Availability Statement:** All relevant data are within the paper, its Supporting Information files,

## Abstract

A large part of our material culture is made of organic materials, and this was likely the case also during prehistory. Amongst this prehistoric organic material culture are textiles and cordages, taking advantage of the flexibility and resistance of plant fibres. While in very exceptional cases and under very favourable circumstances, fragments of baskets and cords have survived and were discovered in late Pleistocene and Holocene archaeological sites, these objects are generally not preserved, especially in tropical regions. We report here indirect evidence of basket/tying material making found on stone tools dating to 39–33,000 BP from Tabon Cave, Palawan Philippines. The distribution of use-wear on these artefacts is the same as the distribution observed on experimental tools used to thin fibres, following a technique that is widespread in the region currently. The goal of this activity is to turn hard plant segments into supple strips suitable as tying material or to weave baskets, traps, and even boats. This study shows early evidence of this practice in Southeast Asia and adds to the growing set of discoveries showing that fibre technology was an integral part of late Pleistocene skillset. This paper also provides a new way to identify supple strips of fibres made of tropical plants in the archaeological record, an organic technology that is otherwise most of the time invisible.

## 1. Introduction

Plants constitute with animals the sources of raw materials of what Hurcombe calls "the missing majority". Indeed, a very large part of the material culture of Prehistoric people was probably, just like ours nowadays, made of organic materials that do not preserve well [1]. Amongst

and the Dryad repository: https://doi.org/10.5061/dryad.2280gb5x9.

**Funding:** The different stages of this project were supported by the European Union's Horizon 2020 research and innovation program under the Marie Sklodowska-Curie grant agreement #843521 (marie-sklodowska-curie-actions.ec.europa.eu), the Institut de Recherche pour le Développement (www.ird.fr), the Muséum National d'Histoire Naturelle of Paris (www.mnhn.fr/fr), Ile-de- France Region (www.iledefrance.fr), the Fondation Fyssen (www.fondationfyssen.fr/fr), the McDonald Institute for Archaeological Research (University of Cambridge) (www.arch.cam.ac.uk/institutes-and-facilities-overview/mcdonald-institute-archaeological-research), the Institute for SE Asian Archaeology (iseaarchaeology.org), and the PrehSEA Program. The funders had no role in study design, data collection and analysis, decision to publish, or preparation of the manuscript.

**Competing interests:** The authors have declared that no competing interests exist.

this prehistoric organic material culture are textiles and cordages, taking advantage of the flexibility and resistance of plant fibres. Fragments of Pleistocene ropes were discovered for the first time in 1953 at the cave of Lascaux by Abbé Glory [2]. Since then, more recent discoveries have pushed the antiquity of fibre technology back in time and the oldest direct evidence to date consists of a fragment of cord made of twisted bark found on a Levallois flake dating to $46 \pm 5$ ka to $52 \pm 2$ ka (MIS 3) and attributed to Neanderthals [3]. Likewise, 30,000-year-old bast fibres of an unknown species have also been found in Dzudzuana Cave, Georgia, associated with fungi and insects feeding on textiles [4, 5], and fragments of 19 300-year-old twisted fibres have been discovered in Ohalo II, Israel [6]. Negative impressions of textiles and basket fragments on fired and unfired clay dating between 27,000 and 24,000 years ago have been uncovered at the sites of Pavlov I and Dolni Vestonice I and II in the Czech Republic [7, 8]. Venus figurines of the Gravettian (27,000 to 20,000 years ago), found across Europe, Ukraine and Russia have been represented wearing woven garments on their bodies and heads, or sometimes with ropes around their wrists which is the case for instance of a large marl figurine and found in Kostienki I [8] (Fig 9), while discoveries of Pleistocene beads imply indirectly that string or sinew technology existed at even earlier times. Intentionally pierced shells were found at the Grotte des Pigeons, Morocco, dating to 82,000 BP, as well as at Blombos, South Africa dating to 75,000 years old [9, 10]. The discovery of naturally perforated shells at the cave of Qafzeh, Israel, pushes the antiquity of string technology even further back as a recent use-wear analysis showed that 120,000 year-old shells were suspended on a string [11–13].

In Southeast Asia, the oldest preserved artefacts made of plant fibres are fragments of mats found in the Lower Yangtze River in Southern China and dated to 8000 to 7000 BP. They are associated with a dug-out canoe, a storage pit filled with fruits and nuts, and pile dwellings [14–16]. Excavations at Niah Caves, Borneo, delivered well-mineralised fragments of baskets, mats, cordages and textiles exceptionally well-preserved in a tropical environment. They were found with Neolithic burials dating to between 3500 and 2200 BP and a detailed study revealed that the baskets were made of rattan and bamboo and the mats of pandan and bamboo [14]. These plants are still being widely used in the region to manufacture similar products [14, 17]. Matting impressions in clay dating to 2750 BP were also found in Lo Gach, Southern Vietnam. A series of indirect evidences complement these finds: Banana trees (*Musa* sp.) micro-remains were found on stone tools from Leang Sarru, Sulawesi, Indonesia dated to 35,000 to 22,000 BP. They were interpreted as possibly related to making rope, basket or woven items as banana fibres are used nowadays to make ropes, baskets and textiles [18, 19]. The finding of a bone fishing gorge dated to before 30,000 BP from Bubog I, Occidental Mindoro, Philippines, provides indirect evidence for the manufacture and use of fishing lines since the Late Pleistocene [20, 21]. *Boehmeria* seeds were recovered from 14,000 to 7000-year-old layers at the sites of Ille and Pasimbahan in Palawan [22, 23]. Today, the bast of Boehmeria is used to make strings, ropes and fabrics and the presence of Boehmeria has been interpreted as evidence for organic technology [24]. Macro-remains of the woody vines *Artabotys*, Mimosaceae, prob. *Entada phaseloides*, and Dilleniaceae cf. *Tetracera* sp. were identified in the Early and Mid-Holocene layers at the site of Bubog II, Occidental Mindoro, in close proximity to Bubog I, all of which have the technological properties to be made into cordage, baskets or even fishing lines [25]. Further East, at the site of Waim in the highlands of Papua New Guinea, an incised volcanic stone probably associated with net-bag making was found in a layer dated to 5050–4200 BP. It is covered in ochre and the V-shape of the groove is characteristic of pulling soft plant fibres. In addition, members of the community near the site who shared traditional knowledge with the involved archaeologists used very similar objects to dye plant fibres to make bilums, the traditional net bags of PNG [26].

In this paper, we report indirect evidence of manufacture of basket or tying materials on three stone tools discovered at the Late Pleistocene site of Tabon, Palawan Philippines identified through use-wear analysis. The appearance and distribution of use-wear on these artefacts is similar to that observed on experimental tools used to thin plant fibres, following a technique that is still widespread in the region and that we recorded in detail among Pala'wan indigenous communities. Our results add to the growing set of discoveries showing that cordage, baskets and mats were an integral part of late Pleistocene material culture in Southeast Asia and elsewhere [14]. To our knowledge, they constitute the oldest evidence of textile and rope technology in the region, together with the banana fibres found on stone tools from Leang Sarru, Indonesia [18].

## 2. Material and methods

### 2.1. Tabon Cave, Palawan, Philippines

Tabon Cave is located on the Southwest coast of the island of Palawan and is one of the major sites of Philippine Prehistory, as it has yielded thousands of lithic artefacts, human bones and some animal, as well as hearth features, with dates ranging from the end of the Pleistocene to the early Holocene [27, 28]. While the cave is now situated just above the shores of Lipuun Point in the municipality of Quezon in Central Palawan, it was located approximately 20 to 30 km inland during the Late Pleistocene, which is also evident in the absence of marine shells in the stratigraphic layers [28, 29].

The site, which is well-known for yielding among the earliest *Homo sapiens* remains found in the region [30, 31], has been the object of renewed investigations in the past decade (see for instance [27, 32–34]; including a closer reassessment of the lithic artefacts.

Human presence in Tabon dates back almost 40,000 years, from the oldest fossils of anatomically modern humans in the Philippines and associated stone tools, to extensive jar burials attributed to the Philippine Metal Age. The cave was excavated in the 1960's by a team from the National Museum of the Philippines and some of the first radiocarbon dates for the region were obtained from the site [28]. More recently, it has been the subject of a reappraisal of the stratigraphy and Uranium series and new AMS $^{14}$C dating (Fig 1) [27, 31–33, 35]. This showed that the early stone tools known as Flake Assemblages II and III correspond to human use of the cave between 39,000- and 33,000-years BP, with an interruption sometime around 38,000 years BP due to increased hydrological activity and possible landscape changes [27]. The new radiocarbon dates were obtained on five charcoal samples collected during the re-excavation of the cave in the 2000s and dated by Beta Analytic and Oxford. The detail of the reassessment of the chronological frame of the site will be the focus of another article [36].

In his discussion of the lithic assemblages from Tabon Cave, Robert Fox defined the "Tabonian" technocomplex which was according to him a long-lasting tradition characterised by the extreme abundance of unretouched flakes [28]. The raw materials of the lithic tools are red radiolarite [38], white chert and andesite. While the presence of cores indicates that lithic tool production was conducted at the site, the limited amount of raw material blocks suggests that the first stage of the reduction sequence took place directly at the raw material source [28]. In the case of radiolarite, the blocks along the nearby Malatgao River, the probable source of supplies located 8-9km from the cave, contain numerous cracks created by intense tectonic stress. As these cracks impact the knapping suitability of the rock, blocks must be tested on the spot, where they are found [38]. In the framework of our experiments (see below), only 5% of the blocks tested were suitable for knapping. The methods used to produce the Tabon assemblages involve direct percussion and encompass discoid, Kombewa and alternating platform system (SSDA) [39–42]. Elongated flakes have been identified too, but these blade-like products

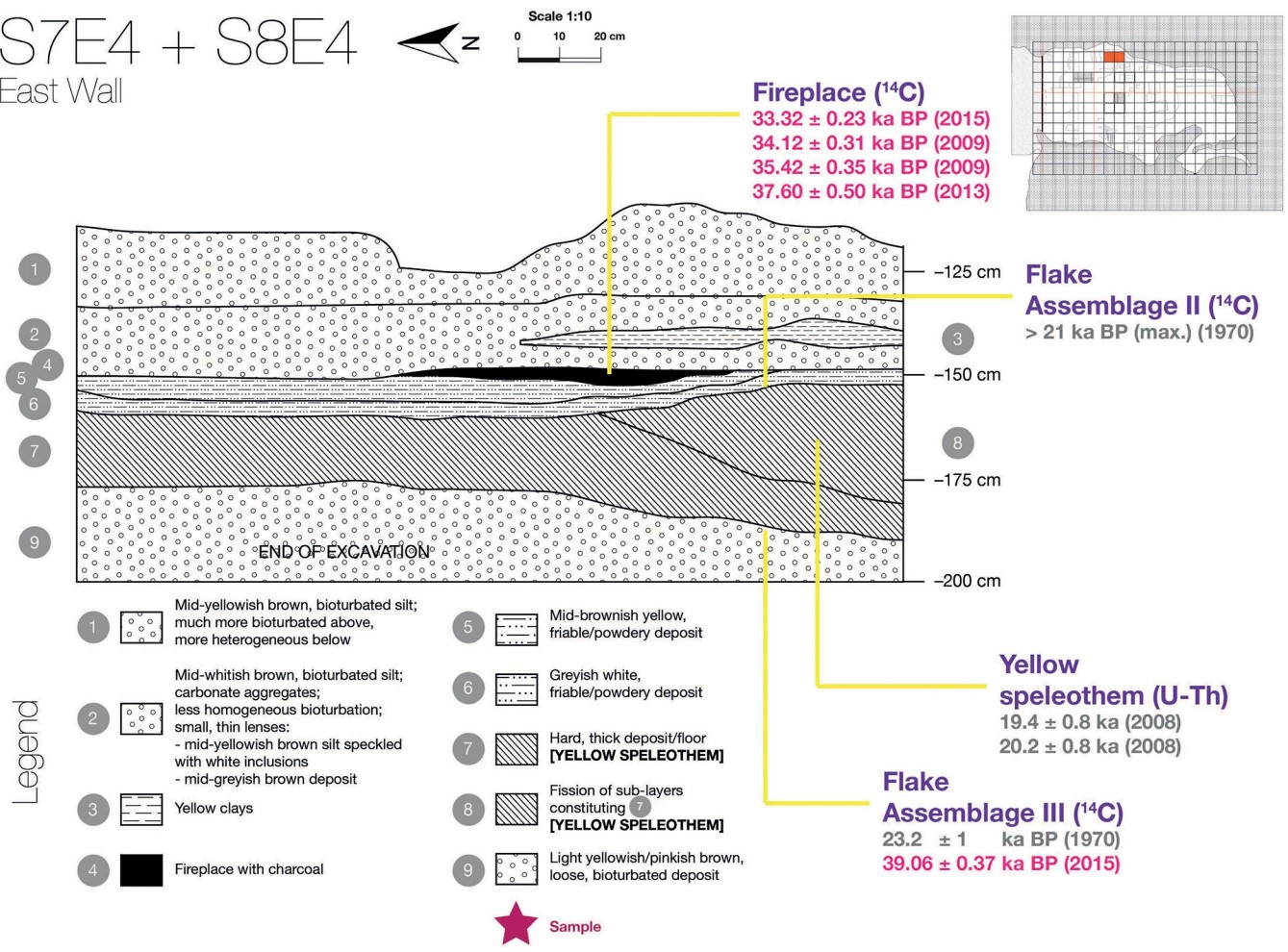

**Fig 1. Location of Assemblages II and III and the associated new $^{14}$C dates from charcoal (in pink).** The new radiocarbon dates are 33.32 ± 0.23 ka BP (BETA-423462), 34.12 ± 0.31 BP (BETA-261802), 35.42 ± 0.35 ka BP (BETA-259326), 37.60 ±0.50 ka BP (OxA-), 39.06 ±0.37 ka BP (BETA-412819). These new dates push the age of the assemblages II and III further back than previously thought based on older dates (in grey) [28, 37]. The detail of the new data on the chronological frame of Tabon Cave are the object of forthcoming publication [27, 36].

would not correspond to actual blades in the technological sense, as they are the result of surface exploitation and not volumetric, the latter implying a construction of the core in three dimensions [42, 43]. Only a few retouched artefacts (8%—scrapers, denticulates, notches) were observed in Flake Assemblages II and III [28, 40], while a large number was found in Flake Assemblage I, the closest to the surface dating to 9,000 BP [28, 41].

The shapes of the unretouched flakes that constitute the majority of the assemblages vary greatly. This results from a lack of standardisation in the knapping process, even in the reduction of a single core. This non-standardisation prevented the establishment of a typology based on morpho-types, and may indicate domestic production [28, 41, 42].

## 2.2. Use-wear analysis of the artefacts

After a preliminary observation of the assemblages with the naked eye and a stereomicroscope, we selected 43 lithic artefacts mainly from Flake Assemblages II and III that seemed to be suitable for further functional analysis because they were not patinated and presented notches resulting from retouch and/or macro-traces due to utilisation. The stone artefacts are curated

at the National Museum of the Philippines and all the necessary permits were obtained for the described study, which complied with all relevant regulations. The accessioning number of the artefacts can be found in S1 File. We analysed them at low magnifications under stereomicroscopes (Olympus SZH and Leica M205) and a macroscope (Leica Z16 APO) (5x to 160x), and at high magnifications using metallographic microscope (Olympus BH2-UMA) (100x to 200x). The tools were cleaned prior to microscope observation in an ultrasonic tank with soapy demineralised water and were then rinsed with demineralised water and left to dry on tissue paper. When needed, alcohol (70–90%) was applied locally during the analysis with impregnated laboratory paper wipes for delicate tasks. The use-wear attributes recorded using a digital data base are outlined in details in S1 File (pp. 8–13) and in a previous publication [44]. Use traces were recorded with digital cameras mounted on the microscopes and marked on photographic illustrations of both faces for each tool. Both stereo-microscope systems were equipped with proprietary image capture software (Leica Application Suite version 4.1.13.0) that permitted automatic stacking of the images to increase the depth of focus. For images taken from the metallographic microscopes, stacking was done using the Helicon Focus software. When residues were present, they were analysed *in situ* on uncoated stone tools, using a Field Emission Gun Scanning Electron Microscope (Quanta-650F) in high vacuum mode.

In this paper we present the results of the analysis of three artefacts from Tabon Cave, Flake Assemblages II and III, dated between 39,000 and 33,000 BP [26], which show diagnostic wear traces that correspond in distribution and morphology to the experimentally generated wear traces that we have observed on experimental tools used to process plant fibres to produce basketry and binding material.

## 2.3. Reference collection: Experiments and ethnoarchaeology

In the framework of an interdisciplinary research program that investigates the prehistoric uses of tropical plants, perishable material culture, and innovations based on vegetal materials, we have been conducting experiments aimed at building a reference collection of use-wear resulting from plant processing, adapted to the vegetation of Southeast Asia [24, 34, 44–46]. To be as realistic as possible, the experiments were based on ethnographic observations of real activities conducted by Pala'wan indigenous people who use wild plants from the rainforest in their everyday life [47]. Before this study began, it was approved by the ethic committees of the Muséum National d'Histoire Naturelle and the National Museum of the Philippines, as well as by the PhD thesis committee of the lead author (HX) H.X. and T.V. were introduced by the ethnographer Nicole Revel to Pala'wan communities who live in the forested foothills and the mountains in the municipality of Brooke's Point, Palawan Island, Philippines. This study was allowed by the councils of elders of the villages of Malia, Makagwaq, Tabud, and Ämrang to whom HX and TV presented the objectives and methods of the research. Consent was given orally by these assemblies. In addition, traditional chieftains, or *panglima*, gave their written consent for this study to take place among their community. Eventually, each participant gave oral approval for being interviewed or filmed systematically in presence of Kristine Joy Colili, who acted as a witness.

We conducted a 3-month fieldwork season in four hamlets, recording information on wild plant processing using tools by means of participatory observation, semi and non-directive interviews, focal person follow (following someone for a half-day or whole day), and spontaneous initiatives from our collaborator-informants (men and women of all ages) [48–50]. H.X. and T.V. recorded activities involving 95 different wild plant species, identified later by J.R.C. and D.T. [24]. Among them, one of the most frequently encountered activities was the processing of plant fibres to make supple strips which would be used to weave baskets and as tying materials. This activity will be detailed below.

Recorded activities, such as making perishable containers using leaves, extracting the hearts of palms, and manufacturing bamboo knives, were then reproduced experimentally by H.X., using tools made of red radiolarite (also called red jasper). These tools were replicas of the artefacts found at Tabon Cave (see [44, 46] for more details). Beyries [51, 52] has shown that the precision of gestures has an incidence on the development of use-wear (e.g. intensity, distribution). In order to experimentally reproduce the activities observed in an ethnographic context as closely as possible, we analysed video recordings of the latter using the concept of *chaîne opératoire*, or operating sequence (see S1 File p.2).

Thinning plant fibres is the final operation in the manufacture of supple strips suitable to make baskets or tying material (Fig 2).

This operation was performed using sixteen experimental tools made of red jasper or radiolarite, a raw material predominant in Tabon Cave and other Late Pleistocene and Early Holocene sites of the area [28, 38, 44]. The tools used were all unretouched flakes obtained by direct percussion with hard hammer, with no or minimal edge torsion. Relatively thin flakes with low edge angle were selected as we thought that these characteristics would allow them to penetrate better between the plant fibres (Table 1).

The experimental tools were used to thin segments of five different plant species: the erect bamboo *Schizostachyum* cf. *lima*, the bamboo vine *Dinochloa luconiae*, the rattan *Calamus merrillii*, the palm *Arenga pinnata*, and *Donax cannaeformis*. The experiments on fresh plants took place in the Makiling Forest Reserve, Luzon Island, Philippines, and we collected parts of the plants to process them once dry in Manila and in France. To avoid interpersonal variability, HX conducted all the experiments. Each operation was conducted twice on a specific plant, first with a first flake for 10 minutes, and then with a second one for 30 minutes (Table 2).

The experimental tools were then analysed using high- and low-power microscopy, following the procedure described above for the archaeological artefacts.

## 3. Results

### 3.1. Use-wear and residues found on 3 stone tools from Tabon Cave

The artefacts P-XIII-T-299, 62-I-11821 and P-XIII-T-59 (Table 3) present a use-wear distribution that is consistent with the one observed on experimental tools used to thin plant fibres. Figs 3–5 show a summary of the use-wear and residues observed on each of the tools, which are displayed with their main active edge upwards. All have in common small notches on one face, resulting from utilisation in the case of P-XIII-T-299 (Fig 3A) and 62-I-11821 (Fig 4A) and from intentional retouch in the case of P-XIII-T-59 (Fig 5D). These notches are associated with parallel striations perpendicular to the edge, as well as micro-polish located on the very edge and locally slightly invasive, expanding likewise perpendicular to the edge (See the distribution of use-wear on the tools in Figs 3–5).

Residues were observed on P-XIII-T-299 (Fig 3E) and P-XIII-T-59 (Fig 5G). On P-XIII-T-299, whitish residues were found next to striations, in parallel to them. Unfortunately, observations under the SEM did not reveal any recognisable structure, aside from mycelium spores (Fig 3F) [53]. However, the spores could indicate indirectly that there were once organic residues underneath. These whitish deposits are located exactly where plant fibres have been in contact with the tool according to the use-wear distribution (see below and Fig 11).

On P-XIII-T-59, residues were observed on a second active edge, which also displayed evidence of retouching (Fig 5F). The residues here were fibrous, and SEM observations revealed that they consisted of elongated volumes that could correspond to the epidermal long cells of Monocots (Fig 5H) (Cutler et al., 2017)—more precisely to elongate psilate (without

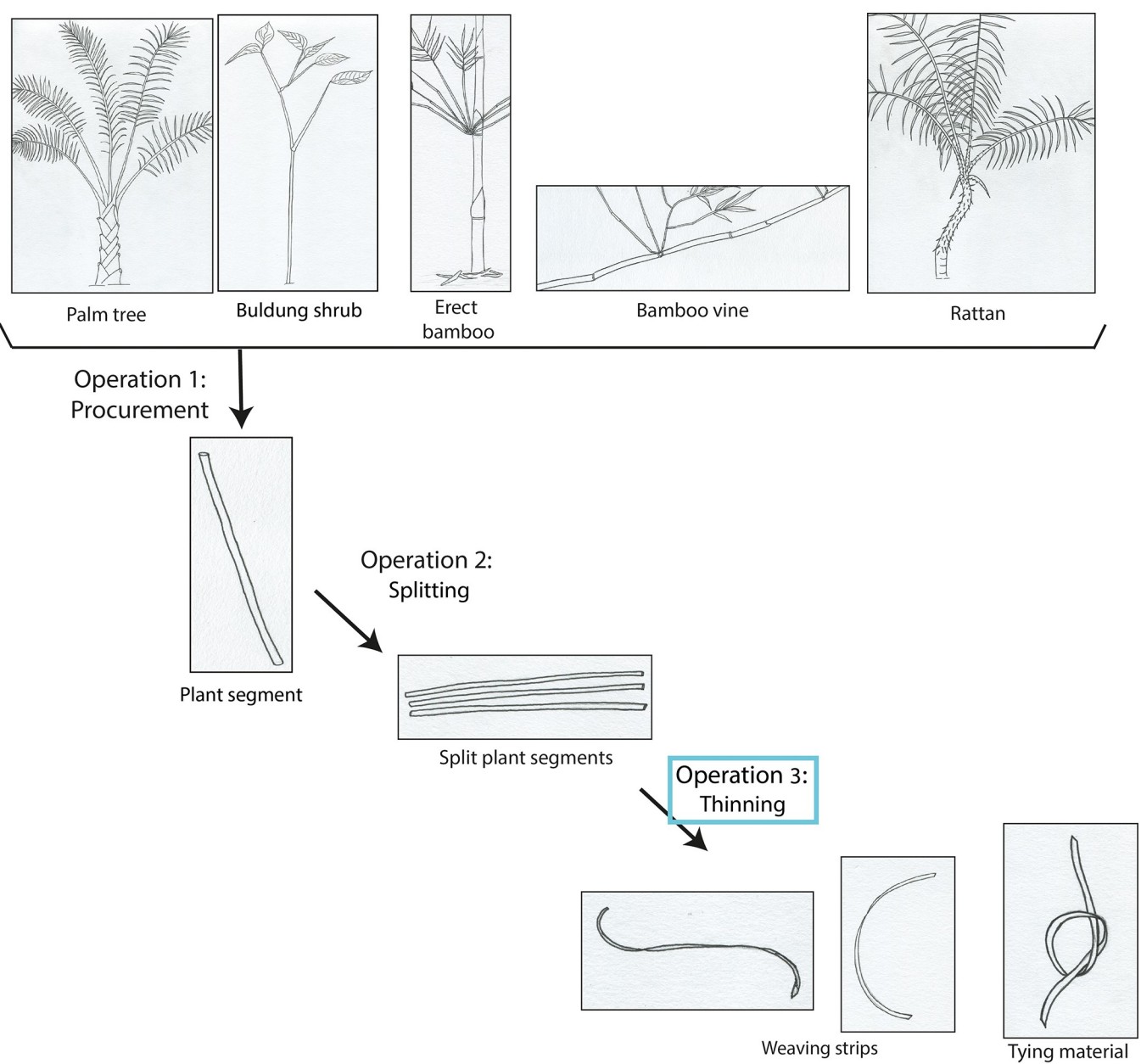

**Fig 2.** *Chaîne opératoire* **of the manufacturing of weaving strips and tying material practiced by Pala'wan communities in Brooke's point municipality, Palawan Island, Philippines.** In addition, segments of bamboo culm and palm leaf petioles are occasionally sanded down, to remove the outer part of the epidermis. The goal is to change the colour of the final product (strips used for weaving) for aesthetic purposes. This is done by scraping off a fine layer of epidermis, using a motion transversal to the tool and longitudinal to the plant fibres. In the case of bamboo, this operation, when performed experimentally with stone tools, creates a very shiny and developed micro-polish as silica accumulates in the epidermis of bamboo [44]. Thinning is the final operation of the *chaîne opératoire*.

ornamentation) epidermal long cells of Poaceae [54]. A comparative study on bamboo [55] demonstrated the possibility of identifying bamboo leaves with long cells, among other anatomical characters (that all silicified into phytoliths). Further analysis would be needed to confirm this possibility, as long cells of similar shape are found in other Poaceae genera such as *Stipa* sp. or *Arundo* sp., as well as other Monocot families such as Posidoniaceae, Juncaceae, and Cyperaceae [56].

**Table 1. Mean and median values, and standard deviation of measurements of the experimental tools.**

| | Technological length (mm) | Technological width (mm) | Max. length (mm) | Max. width (mm) | Max. thick (mm) | Used area length (mm) | Edge angle of used area (°) |
|---|---|---|---|---|---|---|---|
| **Mean** | 40.86 | 37 | 44.69 | 31.63 | 14.63 | 29.65 | 28.34 |
| **Median** | 43.5 | 38 | 45 | 30 | 10.5 | 28.75 | 28 |
| **Population standard deviation** | 10.61 | 11.02 | 7.89 | 7.01 | 11.51 | 10.19 | 7.01 |

Details can be found in S1 File pp.14-46.

Most of the striations observed appeared in a brush-stroke pattern, a diagnostic feature found in abundance on experimental stone tools used to process bamboo [44]. The striations observed on P-XIII-T-59 were particularly similar to the ones observed on the experimental tools used to process bamboo (Fig 5A and 5B), S1 File p. 16, 18, 22, 26, 30, and [44]). The striations being slightly curved on P-XIII-T-299 (Fig 3D) is reminiscent of similar striations observed on experimental tools used to process palm leaf petioles of *Arenga pinnata* [44].

The micro-polish observed along the main active edges of the artefacts (Figs 3, 4B and 5) is not developed enough to be diagnostic when compared to experimental references for the region (e.g. [44, 46, 57, 58]), except on one part of the ventral face of 61-I-11821 (Fig 4D). On this ventral face the polish is flat, shiny, with a high degree of linkage (covering [59]-see the S1 File p.11), cut by dark striations. A similar combination was observed on the very edge of experimental tools used to thin bamboo segments (*Schizostachyum* cf. *lima*) (Fig 7F and S1 File p.46-58). On the second retouched edge of P-XIII-T-59, we recorded the presence of spots of very developed flat, covering and shiny polish on the highest parts of the relief (Fig 5E). Similar polish was observed on experimental tools used to process bamboo, palm, rattan, and *Donax* [44, 46]. Nevertheless, as it is rather isolated here, we cannot completely rule out the possibility that it results from the contact with another stone (bright spot), perhaps with a hard hammer during retouching, or that it corresponds to a taphonomic alteration [60, 61].

## 3.2. Similar use-wear pattern observed on experimental tools used for thinning plant fibres

The use-wear pattern recorded on these three artefacts corresponds to the one observed on the experimental tools that we used to thin plant fibres. Further details of the use-wear observed on the 16 experimental tools used to perform this operation (as well as the terminology used) can be found in the S1 File (P. 14–46). It is summarized below and in Fig 6 and Table 4:

**Micro-scars (Fig 6).** The active edge of the experimental tools used to perform this operation is affected by small to medium micro-scars (100% - 16/16 tools) which are mostly crescent-break (94% - 15/16 tools), probably because of the thinness and low angle of the edges.

**Table 2. List of the thinning experiments performed with unretouched flakes made of red jasper.**

| Plant species | *Schizostachyum* cf. *lima* (erect bamboo) | | | | *Dinochloa luconiae* (bamboo vine) | | | | *Calamus merrillii* (rattan) | | | | *Arenga pinnata* (palm) | | *Donax canniformis* (common Donax) | |
|---|---|---|---|---|---|---|---|---|---|---|---|---|---|---|---|---|
| **Part of the plant** | Culm segments | | | | Culm segments | | | | Stem segments | | | | Segments of leaf petiole | | Stem segments | |
| **Plant freshness** | Fresh | | Dry | | Fresh | | Dry | | Fresh | | Dry | | Fresh | | Fresh | |
| **Duration in min.** | 10 | 30 | 10 | 30 | 10 | 30 | 10 | 30 | 10 | 30 | 10 | 30 | 10 | 30 | 10 | 30 |

**Table 3. Measurements of the three artefacts analysed.**

| | Blank form | Length | Width | Max. thickness | Minimum edge angle | Maximum edge angle | Length of the active edge | Edge profile | Edge form | Edge torsion |
|---|---|---|---|---|---|---|---|---|---|---|
| P-XIII-T-299 | flake | 50 mm | 73 mm | 14 mm | 26˚ | 32˚ | 64 mm | plano-concave | straight | slight |
| 62-I-11821 | flake | 48 mm | 40 mm | 11 mm | 18˚ | 24˚ | 34 mm | plano-concave | concave | none |
| P-XIII-T-59 (edge 1) | shatter | 43 mm | 31 mm | 19 mm | 45˚ | 60˚ | 24 mm | biconcave | concave | none |
| P-XIII-T-59 (edge 2) | shatter | 43 mm | 31 mm | 19 mm | 70˚ | 70˚ | 23 mm | biplane | concave | none |

The length and width of the tools were measures following the technological axis in the case of flakes. In the case of P-XIII-T-59 which is a shatter, it is the maximal length and width that are provided.

They are distributed mostly on the non-contact face in 73% of the cases (11/16 tools) and are oriented perpendicular to the edge in 62,5% (10/16 tools). In three cases, they were localized and formed a notch (19% - 3/16 tools).

**Micropolish (Fig 6).** The very edge often shows a micro-polish which is more developed there than on the rest of the tool (62,5% - 10/16 tools).

**Directional markers (Fig 6).** Different types of directional markers that indicate repeated motions perpendicular or slightly diagonal to the edge (thinning out the fibres) were recorded on all the experimental tools (100% - 16/16 tools).

They are of different types:

—Striations were observed in 81% of the cases (13/16 tools) and are extremely abundant on seven tools (44% - 7/16 tools). They are often long and form large parallel sets (50% - 8/16 tools). Half of the tools showed brush-stroke striations (50% - 8/8 tools), some showed striation-like polishes (37,5% - 6/16 tools), a few showed dark grooves (25% - 4/16 tools) and one tool showed shallow-bright striations (6% - 1/16 tools). The morphology of these striations varies in relation with the processed materials (bamboo, rattan, and *Donax*).

- Polish expanding on the tool surface in perpendicular to the edge was observed on 6 tools (37,5%)

- Stretched-out components forming lines perpendicular to the edge (12,5% - 2/16 tools).

- Fluted polish whose undulations are perpendicular to the edge (12,5% - 2/16 tools).

These directional markers occur predominantly on the contact face of the tool (69% - 11/16 tools).

In addition, some striations parallel to the edge were recorded on 6 tools (37,5% - 6/16 tools). They can be correlated to the marginal sawing motion used to make a first nick to insert the tool between the fibres or pull off a layer of fibres by hand (see below).

The use-wear pattern recorded on the experimental tools used for thinning fibres was not observed on the 73 tools used to perform other tasks, and is thus characteristic of the final operation of the manufacturing of supple plant strips (Fig 7).

The gestures (see S1 File pp. 3–7 for definition of the terms used) performed consisted mainly in a resting percussion transversal to the edge of the tools and longitudinal to the fibres. The angle of contact varied between 0 to 89˚ and the motion was unidirectional. In addition, an occasional bidirectional motion longitudinal to the edge of the tool and perpendicular to the fibres was performed to make a first nick (Fig 8, S1 Video). This was based on the thinning technique used by Pala'wan communities that we recorded in Southern Palawan Island.

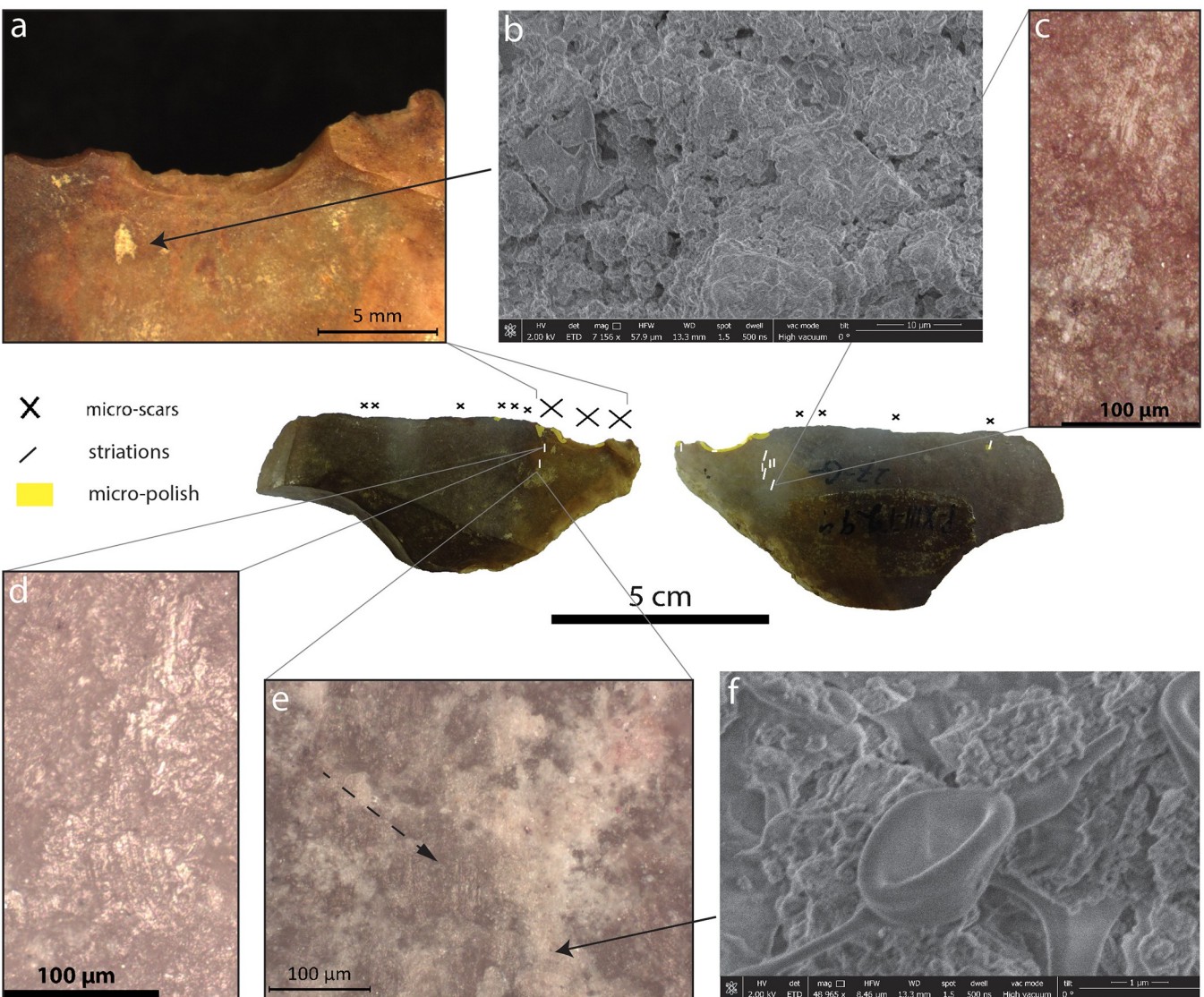

**Fig 3. Use-wear and residues observed on the artefact P-XIII-T-299 from Late Pleistocene layers of Tabon Cave, Palawan, Philippines.** In the centre: both faces of the artefact with use-wear distribution: large micro-scars on the dorsal face, smaller ones on both faces but predominantly the dorsal, micro-polish along the very edge and perpendicular to it, coinciding with the location of the large micro-scars, striations perpendicular to the edge and localised under the large scars. A) large crescent-break micro-scars and light-coloured residue. B) This residue under the SEM. We did not see identifiable structure, except some similar to mycelium (f). C) and D) Brush-stroke striations perpendicular to the edge. E) brush-stroke striations (dashed arrow) partly covered by whitish residues. F) Mycelium [62].

### 3.3. Ethnoarchaeological basis of the experiments: Fibre processing among Pala'wan communities from Brooke's Point, Philippines

Thinning is the final stage of the *chaînes opératoires* of the manufacture of weaving strips and tying material (Fig 2 and S2 Video). It aims at turning rigid plant segments, previously obtained through splitting, into supple strips by removing the inner layers of fibres.

Occasionally, in the forest we witnessed this operation performed with a large machete (tukäw). This was done when an individual summarily wanted to show us how this operation was done. However, on each occasion, they pointed out to us that the appropriate tool would normally be the smaller knife called *paqis* (that is usually kept at home and not carried along in the manner of a machete).

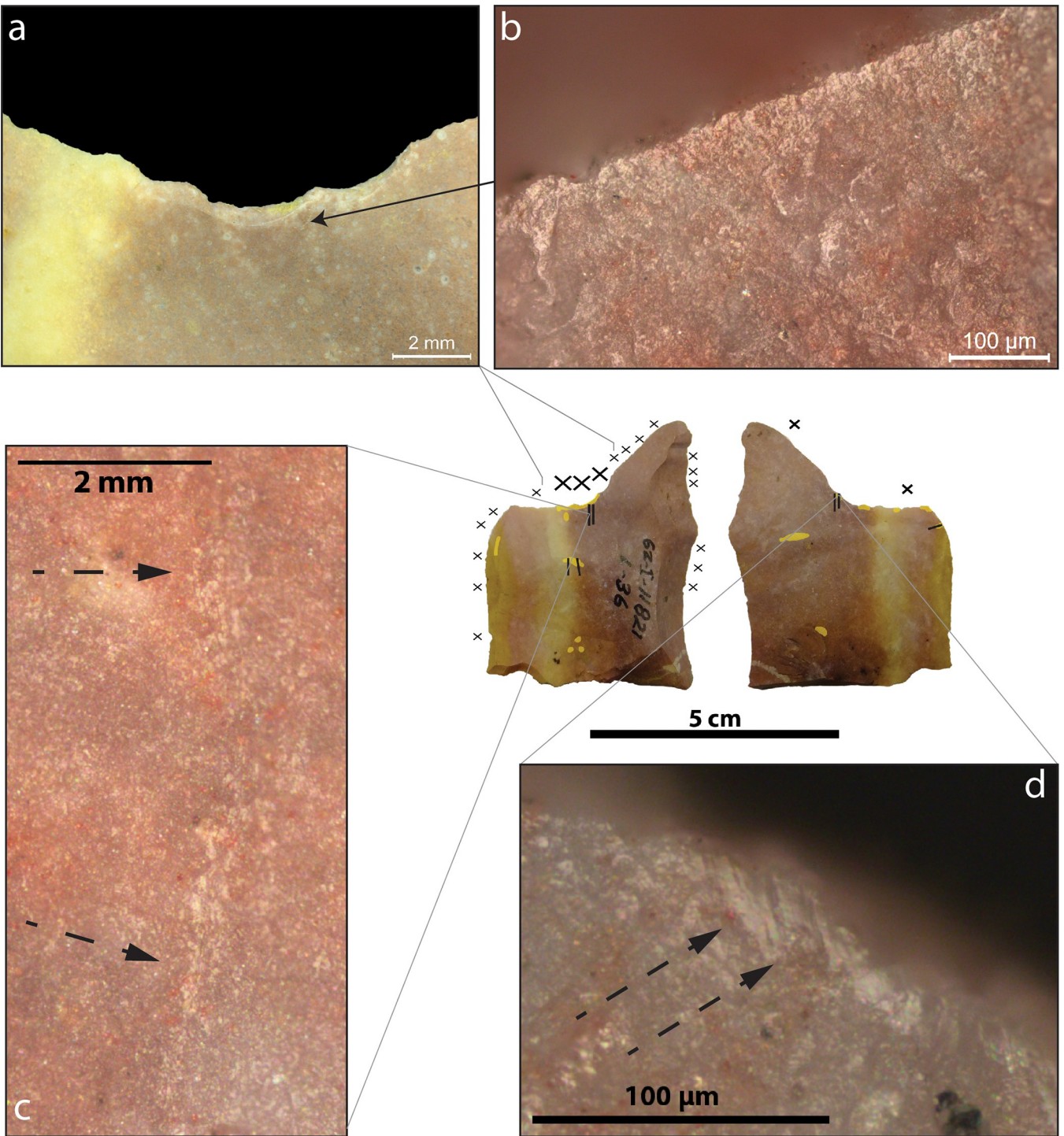

**Fig 4. Use-wear observed on 62-I-11821.** Both faces of the artefact with use-wear distribution: Please refer to the key in Fig 3 for the symbols. A) Concentration of crescent-break micro-scars forming a notch. B) Undeveloped smooth micro-polish along the very edge with some domed components, expanding perpendicularly to the edge. C) Brush-stroke striations that are made of long trails of micro-polish grooved by dark striations. (indicated by dashed arrows). D) Flat smooth micro-polish along the very edge and dark striations (indicated by dashed arrows) perpendicular to the edge.

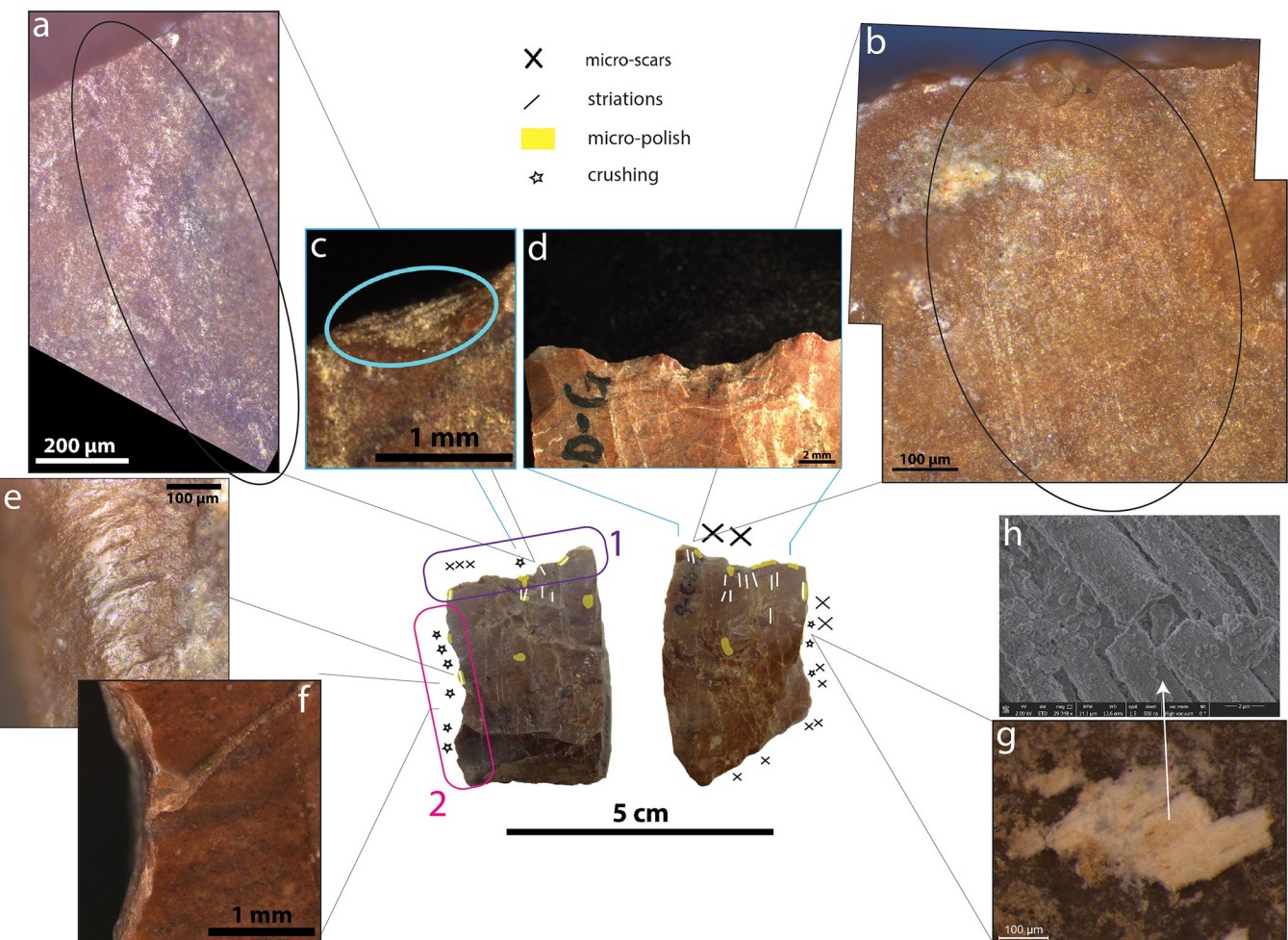

**Fig 5. Use-wear and residues observed on the artefact P-XIII-T-59 from late Pleistocene layers of Tabon Cave, Palawan, Philippines.** In the centre: both faces of the artefact with use-wear distribution. Two edges have been retouched: 1 and 2. Edge 1 presents two notches with counter-bulb (d) and crushing (c) typical of retouch with hard hammer (see [34]). These notches are on the lower-face (right picture of the artefact which is a shatter) and are associated with smaller scars and numerous striations perpendicular to Edge 1 (a and b). Edge 2 has been retouched all along by making abrupt notches whose edges are crushed, indicating a contact with a hammer (f). This series of notches gives the edge a concave shape, similar to the one reported by Fuentes and colleagues [18] on artefacts from Leang Sarru, North Sulawesi. On this edge, we recorded the presence of fibrous residues (g). Based on their shape at high magnification (h), these could correspond to long cells arranged in longitudinal files and resembling Monocots leaf epidermis arrangement ([63] p.79-81). Moreover, elongate psilate epidermal long cells are found to be diagnostic of Poaceae [54] and can even characterize bamboo leaves to the genus level [55]. At one place of this edge, we observed a very developed flat, covering and shiny polish (e).

A major part of the time—sometimes *all* the time—is devoted to a repeated resting percussion with a motion perpendicular to the edge of the tool and longitudinal to the fibres of the plants. The small knife is held between the ring finger and the thumb of the right hand. The blade is slid along the inner face of a narrow plant segment to remove the inner fibres. A downward pressure is applied by the thumb on the blade and an upward counter-pressure on the plant by the index and/or the middle finger of the same hand, tightening up the plant segment. The other hand may help in the action by pulling the plant strip backwards while the blade runs forwards in the opposite direction. This motion is repeated until the segment becomes flexible all along its length, transforming it in a supple strip which is suitable for tying (called *nawiq* and made from *Calamus* spp. and *Lygodium circinatum*) or for weaving baskets (called *pawäk* and made from *Schizostachyum* sp., *Dinochloa* sp., *Arenga* spp., *Donax*

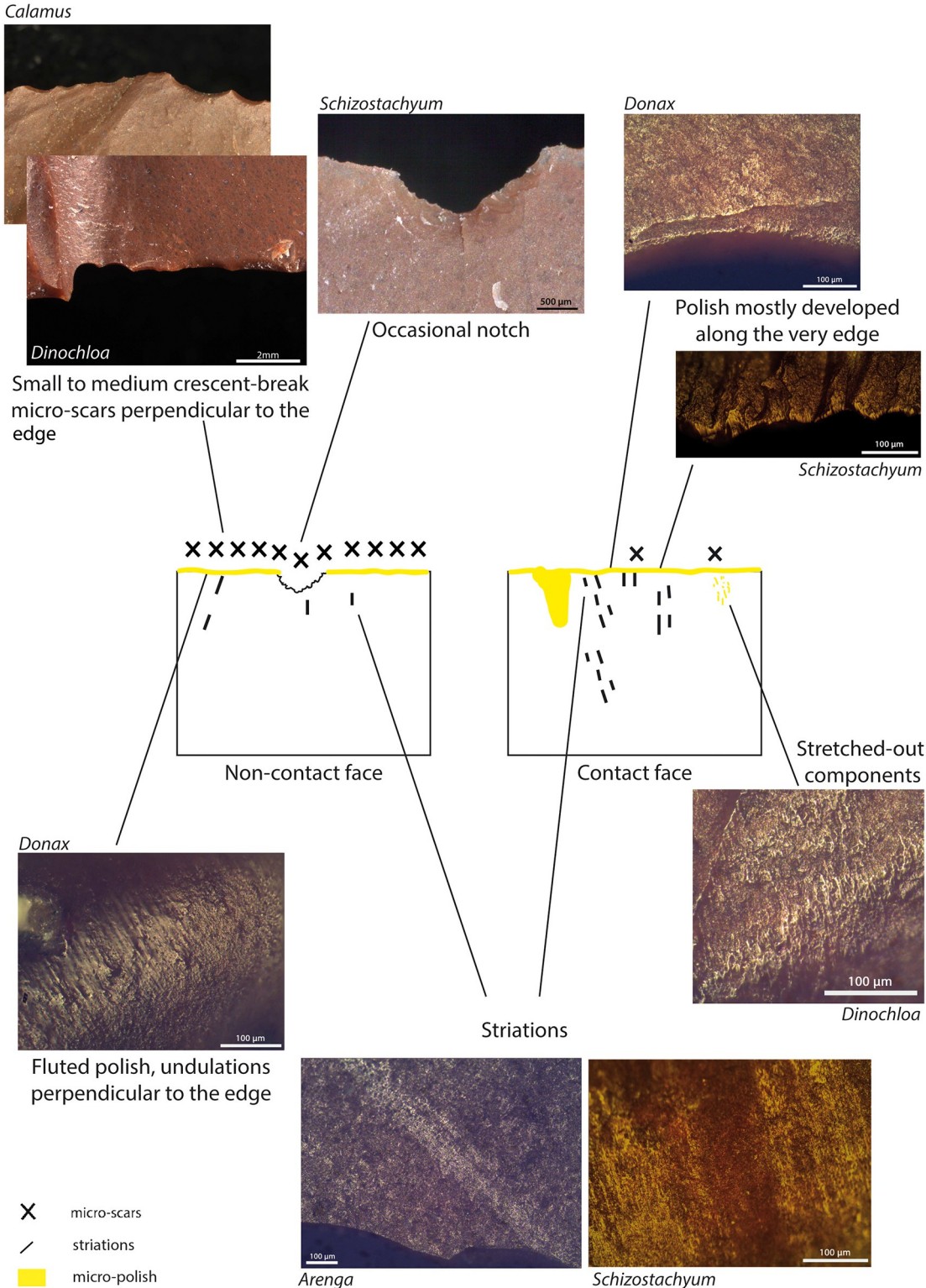

**Fig 6. Schematic representation of the pattern of use-traces that was recurrently observed on the experimental tools used to thin plant fibres, turning rigid segments into supple strips.** All the features are here grouped into one drawing, but they do not necessarily all occur on the same lithic implement (see S1 File p.14-46 for the details). The pictures show examples of characteristic attributes observed on the experimental tools. The genera of the plants processed with them are indicated in italics.

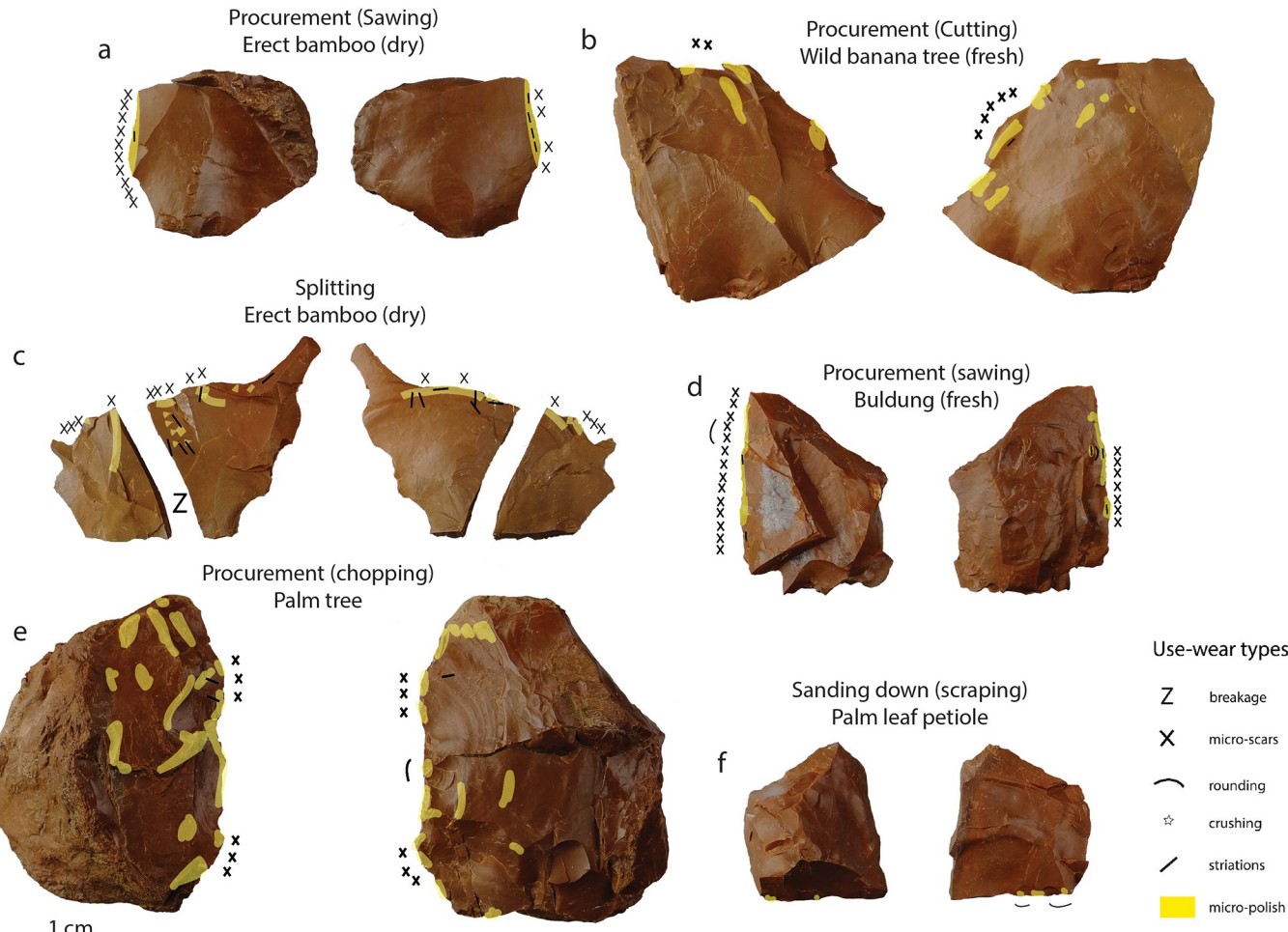

**Fig 7. Examples of use-wear distribution on experimental tools used for tasks other than thinning plant fibres.** a) Procurement of *Schizostachyum* cf. *lima* (culm), b) Procurement of *Musa* sp. (pseudostem), c) Splitting *Schizostachyum* cf. *lima* (culm), d) Procurement of *Donax cannaeformis* (stem), e) Procurement of *Caryota rumphiana* (pseudo-trunk), f) Sanding down (scraping) the epidermis of *Arenga pinnata* (palm leaf petiole). (For more details, see [46]).

*canniformis* and *Calamus* spp.) (Fig 8A and S2 Video— https://youtu.be/qiezpMVvQlI). Additionally, the same gesture can be used to shave the sides of the plant strip (Fig 8B and S2 Video— https://youtu.be/qiezpMVvQlI). A little nick may also be made at one tip of the plant segment and a layer of fibres detached by hands (Fig 8C and S2 Video— https://youtu.be/qiezpMVvQlI). Videos uploaded on www.plantuseinseasia.net show this operation recorded among Pala'wan communities within a wider context as well as its experimental reproduction with stone tools.

While thinning plant segments using metal blades allows one to strip away relatively thick layers of fibres, doing the same gestures with stone tools that are less sharp and have a less acute edge angle removed less fibres at once and the process took longer. Stone tools were nevertheless efficient and able to perform the task successfully. *Calamus* was the hardest plant to work, as its fibres stick strongly together. This was mentioned by our Pala'wan informants using metal knives, and was also experienced by H.X. using stone tools. Rattan was easier to process when dry.

*During our stay, basket weaving was always performed by women.* Women also gathered the raw materials (plants) in the forest and processed them into supple strips. In contrast to this,

**Table 4. Use-wear observed on experimental tools used to thin segments of plant fibres to turn them into supple strips.**

| Micro-scars | Micro-polish | Directional markers |
|---|---|---|
| •Small to medium<br>•Crescent-break<br>•Perpendicular to the edge<br>•On the non-contact face<br>•Occasionally localized and forming a notch | •Most developed along the very edge | Indicating repeated motions perpendicular or lightly diagonal to the edge (thinning):<br>•Striations which may be long and form large sets parallel between them<br>•Undulations of fluted polish<br>•Stretched-out polish components forming lines perpendicular to the edge<br>•Polish expanding perpendicularly to the edge<br>More marginally, striations parallel to the edge, testimonies of the longitudinal motion performed to make a first nick to insert the tool between fibres or to pull a layer of fibres off by hand. |

we recorded the production and use of tying materials, cords, or ropes, by both women and men. Though making weaving strips is a female activity for the Pala'wan communities we stayed with [47], men also knew how to process the plant segments for weaving strips since the same chaîne opératoire was used to make ties. For example, the person who showed us how to process the bamboo vine *Dinochloa* sp. and *Donax canniformis* was a man: Linggit Rilla.

We recorded the use of bamboo culms (erect and vine), stems of *Donax canniformis* and leaf petiole of palms to make weaving strips for basketry and traps. Tying materials were made from rattan stems and the stem-like leaves of the creeping fern *Lygodium* (Fig 9). Weaving strips were used to make basket-containers of different shapes, rice winnowers, and traps. Tying materials made of rattan were used to assemble and hold together the different pieces of houses and objects, while strips of the fern *Lygodium* were used to hold together smaller objects or small parts of larger composite artefacts (Fig 10).

## 3.4. Developing an understanding of the function of Tabon Cave artefacts based on experimental and ethnographic data

The distribution of the use-wear on the artefact P-XIII-T-299 strongly suggests that the tool was held using the left hand (Fig 11). The experimental data showed that thinning plant segments produces micro-scars mainly on the face opposite to the one in contact with the strips (Fig 6, Table 4). In addition, this operation resulted in the appearance of striations mainly localised on the face in contact with the plants. Given the asymmetric morphology of the artefact P-XIII-T-299 and the fact that large scars are localised on the upper-face of the tool and the striations on the lower-face, we can hypothesise that it was manipulated using the left hand. This is in line with the observation of Jensen [59] who interpreted similar use-wear distributions on Mesolithic and Neolithic tools from Denmark as a result of the processing of plant fibres by left-handed people.

The morphology of two notches (Fig 5C and 5D) on the artefact P-XIII-T-59 shows that they result from intentional retouching [34]. Notches on the active edge are very useful to maximise the efficiency of stone tools to thin plant segments. During the experiments, HX took advantage of little concavities in the edge (sometimes created by micro-scars) into which plant strips could be wedged while thinning them down. In several instances, these concavities became more and more affected with each stroke and eventually formed a notch (Fig 6).

Finally, two elements converge to suggest that the active edge 2 of the tool P-XIII-T-59 could have been used to scrape the epidermis of a plant. The first is the micro-polish identical to the one observed on experimental tools used to scrape the epidermis of the bamboos *Schizostachyum* and *Dinochloa*, and the leaf petioles of the palm *Arenga* [44, 46]. The second element is the presence of phytoliths which correspond to the epidermal cells of Monocotyledons. Scraping the epidermis of plants is practiced nowadays by Pala'wan artisans to change the colour of the plant segments in basketry and to remove the itchiness it causes in the case of the bamboo *Schizostachyum*. Edge 2 of the artefact P-XIII-T-59 was intentionally

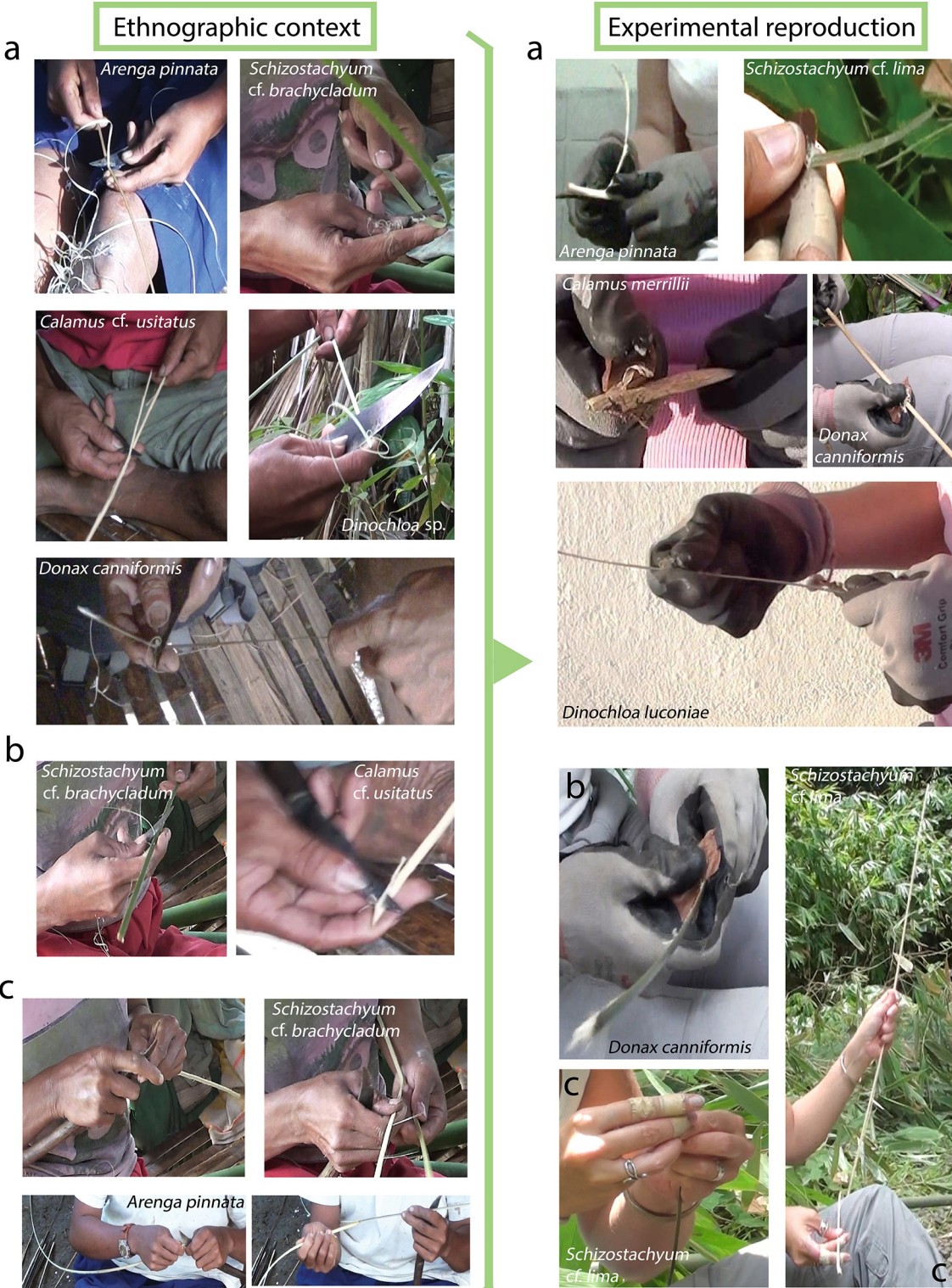

**Fig 8. The thinning operation recorded among Pala'wan communities and its experimental reproduction with stone tools.** It aims at turning rigid plant segments into supple strips suitable for weaving and tying by removing the inner fibres. a) Most of the time is devoted to a resting percussion with a motion perpendicular to the edge of the tool and longitudinal to the fibres. b) Occasionally the same gesture is used to shave the sides of the plant strip. c) A little nick is sometimes made at one tip of the plant segment and a layer of fibres detached by hands.

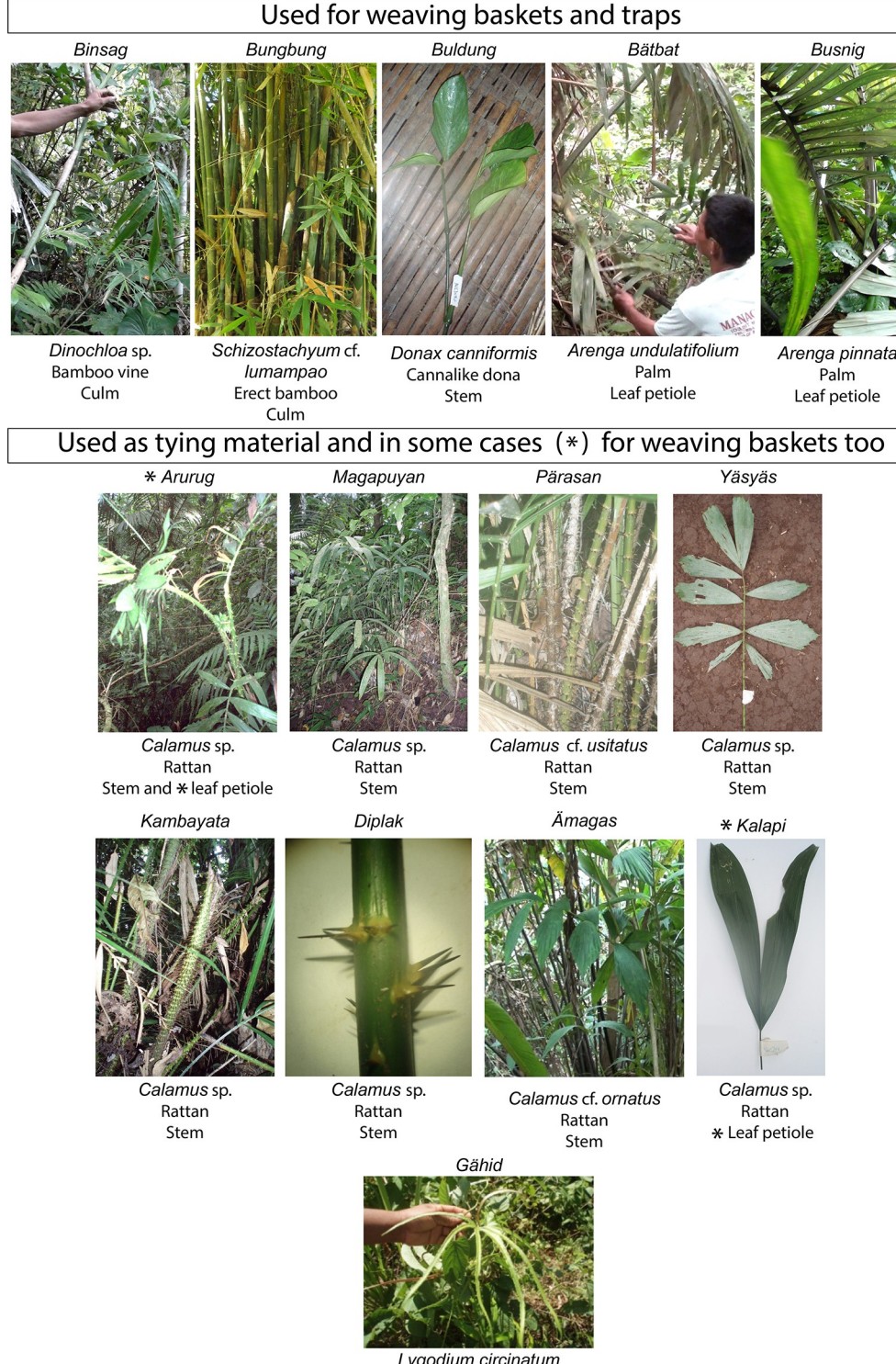

**Fig 9. Plant taxa thinned by our Pala'wan informants to make supple strips suitable for weaving baskets and traps and to use as tying material.**

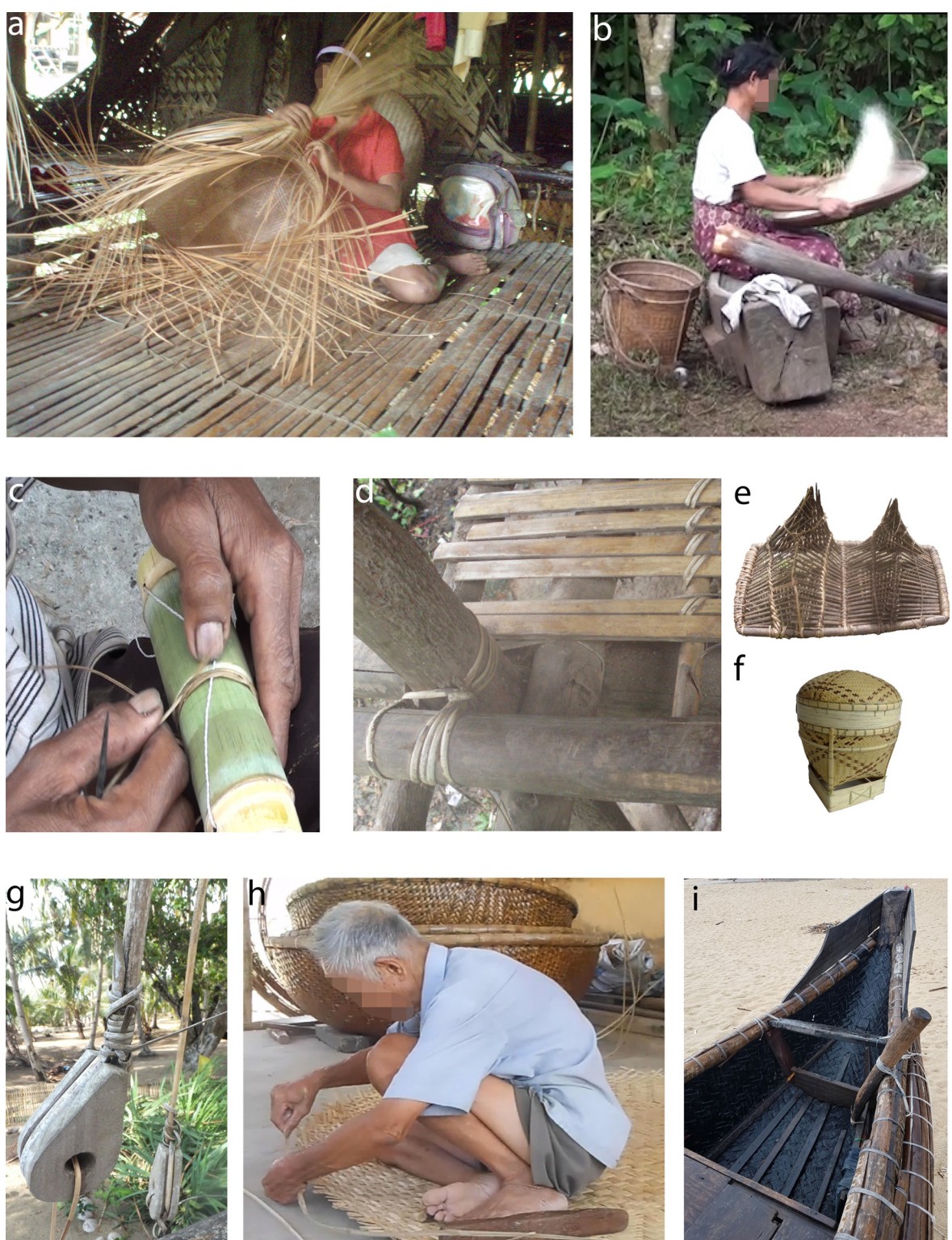

**Fig 10. Uses of weaving strips and tying materials obtained by the *chaîne opératoire* described in this article.** a) to f) were recorded among Pala'wan communities in Brooke's Point municipality, Palawan Island. a) Inin is weaving a large basket (*tabig)* using strips made of *busnig* (the palm *Arenga pinnata*–leaf petiole), and *buldung* (Comon Donax: *Donax canniformis*—stem). Hamlet of Ämrang. b) Bisina is removing the husk from rice grains using a woven winnower. Beside her is a *tabig* to carry the rice. Hamlet of Malia. c) Maniwang is attaching together the different parts of the double bamboo container *aläp*. Hamlet of Ämrang. d) Post, beams, and flooring of a house.

The different parts are attached together using rattan stems. Hamlet of Ämrang. e) Fish trap made of strips (palm *Arenga undulatifolia)* attached together by ties made of rattan. On the way to Ämrang coming from Mäkägwaq. f) Small *tingkäp,* a basket produced mainly for commercial purposes. It is woven using bamboo, palm, and common Donax strips. The different parts (rim, base, etc.) are assembled using the fern *gähid* (*Lygodium circinatum*) [64]. g) to i): Use of plant strips for navigation. g) Ropes made of rattan on a Cuyonin boat, Ecomuseum of Sibaltan, Northern Palawan. h) An 87-year-old craftsman is making a boat by weaving bamboo strips, village of Kim Bong, Vietnam. Resin is used to waterproof the boats and they can last up to 5 years. i) Boat of a different morphology made by weaving, in Thon Thai, Duong Ha, Hue, Vietnam.

retouched as well, giving it a concave shape, perfectly suited to processing a convex contact material such as a bamboo culm or a palm leaf petiole. For all these reasons, we cannot rule out that this artefact was used for several steps of the *chaîne opératoire* of basket making: scraping the epidermis, and thinning plant segments, to turn them into supple strips. (See S1 File p.46-58).

## 4. Discussion

Diagnostic use-wear pattern characteristic of thinning plant fibres were determined on three stone tools from Tabon Cave dating back 39–33,000 years. We were able to identify this pattern because it is identical to the use-wear distribution observed on experimental flakes used to process rigid plant segments, turning them into supple strips suitable for weaving or as tying materials. The use-wear pattern is characterised by a specific distribution of striations, micropolish, and micro-scars on the surface of stone tools (Figs 3–6): (1st) The active edge is affected by small to medium micro-scars located in most cases on the face that was the least in contact with the plant (the non-contact face). They tend to be oriented perpendicular to the edge and are occasionally localised, forming a notch. (2nd) The very edge shows a micro-polish which is more developed than elsewhere on the tool. (3rd) Mainly the contact face but sometimes also the non-contact face exhibits directional markers in perpendicular and diagonal orientation to the edge which are often numerous and indicate a repeated transversal motion. These directional markers consist of striations of various kinds which are often long and parallel to each other, and can form larger sets parallel between them or long trails that materialise the contact with the processed plant strips, undulations of fluted polish, stretched-out polish components forming lines, and/or polish expanding inside the tool surface (Fig 6).

Identifying plant thinning based on the identification of specific use-wear patterns has been done in Europe for periods ranging from the Mesolithic to the metal ages [65, 66]. To our

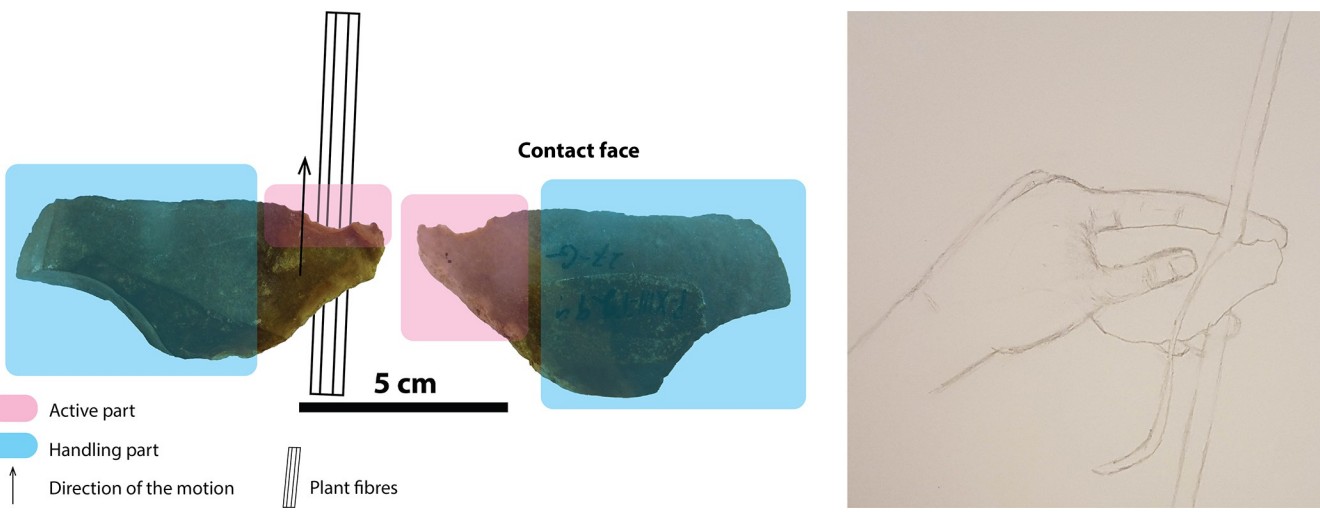

**Contact face**

Active part

Handling part

Direction of the motion

Plant fibres

5 cm

**Fig 11. Reconstruction of the use of the artefact P-XIII-T-299 from Tabon Cave.**

knowledge, our study is the first to present in detail a use-wear pattern resulting from thinning segments of tropical plant taxa (see [57] for a preliminary description of the traces resulting from processing rattan).

For many decades, researchers working in Southeast Asia have formulated the hypothesis that simple and non-standardised lithic artefacts had been complemented by plant implements made of bamboo [44, 67–71]. In contrast, our results constitute evidence for the existence of perishable plant-based technology in prehistoric Palawan, even though the number of archaeological artefacts displaying this thinning pattern is still limited. They show that early plant technology did indeed exist in tropical regions during the Pleistocene, but it was not focused simply on the manufacturing of bamboo tools. This complements our discovery that some of the denticulates from Tabon Cave had been used to split plants. Split plants form component parts of many objects nowadays, including flooring, musical instruments, and darts [34]. It is also in line with the results of Riczar Fuentes and colleagues [18] who found residues of banana tree fibres on artefacts from Sulawesi, Indonesia, considered by the authors as evidence for making ropes, baskets, traps, or other woven items, and the use-wear traces observed by Elspeth Hayes and her colleagues on 18–11,000 year old artefacts from Liang Bua, Flores Island, Indonesia, which they interpreted as related to fibrecraft [58].

Establishing experimental use-wear reference collections created as a result of realistic activities based on observations of skilled craft makers in ethnographic contexts, offers a powerful tool to recover evidence of perishable material culture, as we demonstrated here and as advocated elsewhere [51, 52, 72, 73]. The technique we described to thin plant fibres is widespread nowadays and has been reported among other indigenous groups in Palawan [74], but also in areas more distant from the island such as Northern Luzon [57], the Batanes [19], Borneo [75], Southern China [71], and Lao (Davenport, pers. com.). Aside from the occurrence of the same thinning technique in regions separated by hundreds and even thousands of kilometres, some decorative motifs such as the one called "pigeon's eye" are also shared by different groups in the region [75]. Sellato [75] suggested that these similarities of style and raw materials used may derive from an old common cultural stock. We fully agree with him and add a common processing technique to the equation, one already practiced as early as 39–33,000 years ago.

The identification of plant thinning in the archaeological record of Tabon Cave through our microscopic use-wear analysis has several implications. It shows that traceology can be used to make organic technology visible in the archaeological record, even if the perishable final products themselves were not preserved. The results of our study suggests, furthermore, that the technology of processing plant fibres and manufacture of cords, weavings, and basketry already existed 39–33,000 years ago in Southern Palawan. The supple plant strips obtained may have been used as tying materials, to make baskets or traps, as Pala'wan communities do nowadays, or they could have been used for other purposes. Strings and cords can be used to hold ornaments like beads or shells, something that has been practised by our species for 120,000 years [76], and for complex technologies such as bow hunting. Langley and her colleagues showed that bone points had been hafted and shot using bows as early as 48,000 years ago in Sri Lanka [77]. Today, the mouth of Tabon Cave is overlooking the sea, but during the Late Pleistocene, it was c. 30–35 kilometres further inland [28, 29]. It is nevertheless worth noting that the manufacture of ropes, nets and wickerwork are technologies fundamental for navigation, and more generally for populations in maritime environments ([78], Fig 9). The palm *Arenga* sp. mentioned in this study has been identified as the raw material for the ropes of the San Isidoro shipwreck dating to the 19th century and found along the Western coast of Luzon Island, Philippines [79], while *Caryota mitis* was used to make sailing ropes in the same country at the beginning of the 20th century [80]. Plant strips can also be used to manufacture entire boats, a practice extant today in Vietnam (Fig 9H to 9I) [81, 82]; and fibre technology

was necessary for fishing using hooks and nets. Archaeological evidence for these fishing practices has been found from the Late Pleistocene onwards on the islands of Mindoro, Alor, and Timor, at the sites of Bubog 1 and 2, Tron Bon Lei, and Asitau Kuru (formerly Jerimalai) [78, 83–90]. Kerfant [19] highlighted the interesting aspect that most of the plants used in seafaring come from forests away from the coast, implying the mastery of both marine and inland environments. This is further indicated by the remains of forest dwelling animals like deer, pigs, and cloud rats found together with marine vertebrate and invertebrate remains in coastal sites [21, 90–92]. Ties made of rattan were used until recently to attach shafts to stone tools in Papua New Guinea [93, 94], and may have played an important role in hafting for a very long time. More generally, tying materials enabled the development of composite technologies [95]. With the use-wear pattern presented here, we now have a means to make fibre technology more visible in the archaeological record, even if the perishable final products have not survived.

It is worth noting that the pseudo-stem of banana trees (mainly *Musa textilis*) is currently used to make fabrics, while the leaves of pandan are processed and woven to make mats [19, 24, 46]. We also conducted experiments with these materials, but they created traces which differed in distribution and morphology from those we reported here. This seems to be related to the fact that 1) the gestures involved are different, and 2) the physical properties of these plants are also different. Silica and water content influences the degree of hardness abrasion and the formation of use-polish [44, 45].

Stone tools found in Southeast Asia generally derived from simple production techniques that persisted with very few changes for thousands of years [68, 69, 96–98]. This situation led some prehistorians to dismiss the archaeology of the region and to assume that it did not play a significant role in technological innovations [99, 100]. In line with other studies showing for instance that the earliest representation of a scene is from Sulawesi, or that people had a mastery of navigation (e.g. [18, 85, 86, 101, 102]), our results contribute to rehabilitate Southeast Asian prehistoric heritage by showing that prehistoric groups had in fact great technological expertise and advanced skills, including the ones necessary for thinning plant fibres [75] and possibly for assembling strips to manufacture three dimensional objects. In fact, the human groups who inhabited Tabon Cave had developed a botanical knowledge deep enough to know which plants within their environment had fibrous, flexible, and solid properties, and could be turned into ropes, baskets, and other fibrecraft [75]. Monocotyledons such as bamboo and palms are particularly appropriate for these applications because phytoliths, or biogenic silicates, precipitate in their epidermis, giving them strength [19, 103, 104].

As early as 39–33,000 years ago, the Late Pleistocene inhabitants of Palawan possessed an elaborate organic technology and were processing plant fibres to make cordage, baskets, traps, or other composite objects. More than 30,000 years later, this botanical knowledge and technological know-how are still alive and allow many communities all over Southeast Asia to produce objects necessary to answer their everyday needs in a sustainable way. It remains to be explored whether these contemporary practices are the fruits of a continuous tradition directly rooted in the Late Pleistocene. It is quite possible, as Dario Novellino [74] has shown, that knowledge of plant resources and technological skills tend to be more resilient and transmitted to other generations than traditional narratives or cosmogony.

## 5. Conclusion

In this paper, we reported the discovery of a use-wear pattern characteristic of thinning plant fibres found on three stone tools from Late Pleistocene deposits at Tabon Cave, Palawan Island, Philippines, dating to 39–33,000 BP. These were identified based on a comparison with

the distribution of use traces on experimental flakes used to turn rigid plant segments into supple strips, reproducing as closely as possible a technique that we recorded in ethnographic contexts among Pala'wan communities, and that is widely distributed in the region nowadays. These results show that seemingly simple Southeast Asian artefacts are hiding testimonies of a behavioural complexity invisible to the naked eye but that we can reveal through use-wear analysis. It had been hypothesized by proponents of the Bamboo Hypothesis that Southeast Asian stone tools had been used mainly to manufacture bamboo tools and that the focus of prehistoric craft makers on this giant grass (Bamboo is a Poaceae) would explain the technological simplicity of lithic artefacts [67–70]. Our results add to the recent evidence showing that a plant-based perishable technology indeed existed in the region during Prehistory, but also add nuance to the Bamboo Hypothesis *sensu stricto*, showing that people invested in plant materials in a much broader sense and did not use their stone tools exclusively to make bamboo knives, arrows and darts. Among this perishable technology were tying materials and possibly baskets, traps, or other objects resulting from the weaving of plant strips such as the ones made using the artefacts from Tabon Cave. This discovery highlights both the technological skills and botanical knowledge possessed by the inhabitants of Southern Palawan at the end of the Pleistocene. So far, these results constitute among the oldest evidence for fibre technology in the region, together with residue and use-wear analyses from Leang Sarru in Sulawesi [18]. Future analyses of lithic artefacts from Southeast Asia using the experimental reference we provide here can reveal if the antiquity of fibre technology was even greater than 39–33,000 BP in this region, the geographical extent of this craft, and if its contemporary practice is the result of an uninterrupted tradition.

## Supporting information

**S1 File.**
(PDF)

**S1 Video. Experimental thinning of plant fibres using stone tools.**
(3GP)

**S2 Video. Plant fibre processing by members of the Pala'wan community from Brooke's Point, Philippines.**
(3GP)

## Acknowledgments

We would like to dedicate this article to Sheldon Jago-on who left this world too soon. May his passion for Tabon Cave and lithic artefacts inspire us for many years. As he passed away before the submission of the final version of this manuscript, Hermine Xhauflair accepts responsibility for the integrity and validity of the data collected and analyzed.

We are very grateful to the National Museum of the Philippines for allowing us to conduct this research and giving us access to the archaeological artefacts from Tabon. Our study was conducted with the informed consent of the Pala'wan communities involved, who also authorised us to publish the data. We thank warmly our Pala'wan collaborator-informants for welcoming us so well and for accepting to share some of their knowledge with us. We are grateful to J. Colili, our interpreter, N. Colili whose help made this work possible, and N. Revel for introducing us to Pala'wan communities as well as for her consistent and supportive guidance through the years. We thank the botanists who helped to identify the plant taxa involved: R. Bandong and J. LaFrankie of the Department of Biology of the University of the Philippines, Diliman, N. Pampolina, P. Malbrigo, D. M. Pulan, the Department of Forest Biological

Science, University of the Philippine Los Baños and Khoon Meng Wong from the Singapore Herbarium. We thank the Makiling Center for Mountain Ecosystem for permission to collect plants and conduct experiments and the National Museum for their authorization to analyse the artefacts from Tabon Cave. Thanks to V. Zeitoun, V. Paz, H. Lewis, G. Lucarini, and the UMR 7206 for their help with logistics, to R. Fuentes and A. Tiauzon for their collaboration in knapping the experimental tools and to E. Turpin, E. Robles and L. Manta-Khaira for helping to record the experiments. We are deeply grateful to S. Beyries and her colleagues from the CEPAM for welcoming HX very nicely at the Université Côte d'Azur in their laboratory of use-wear analysis. In Cambridge, we were assisted most kindly by G. Lampronti and I. Buisman to conduct SEM analyses and received encouragements from J. Stargardt, G. Barker, and D. Gaffney. We would like to thank M. Tromp and N. Wills for their help with the micro-plant remains, and J. Ibañez for his comments on an earlier stage of this manuscript. The presentation of the new dates from Tabon was possible thanks to the efforts of the National Museum of the Philippines and the MNHN in the frame of the PrehSEA programme, in particular F. Sémah, E. Dizon, X. Gallet, F. Détroit, and J. Corny. Eventually, we are immensely grateful to V. Woué, O. Nanga, R. De Coster, G. Xhauflair, and R. Morgan for their great interest in this research and never-failing support, and to the editor, the two anonymous reviewers, and the wonderful proof-reader, for their encouraging and constructive comments that helped us improve this article.

## Author Contributions

**Conceptualization:** Hermine Xhauflair.

**Data curation:** Hermine Xhauflair.

**Formal analysis:** Hermine Xhauflair, John Rey Callado, Danilo Tandang, Céline Kerfant.

**Funding acquisition:** Hermine Xhauflair.

**Investigation:** Hermine Xhauflair, Sheldon Jago-on, Timothy James Vitales, Dante Manipon, Noel Amano, John Rey Callado, Danilo Tandang, Céline Kerfant, Omar Choa.

**Methodology:** Hermine Xhauflair.

**Project administration:** Hermine Xhauflair, Sheldon Jago-on, Timothy James Vitales.

**Resources:** Hermine Xhauflair, Sheldon Jago-on.

**Supervision:** Alfred Pawlik.

**Validation:** Alfred Pawlik.

**Visualization:** Hermine Xhauflair, Dante Manipon.

**Writing – original draft:** Hermine Xhauflair, Dante Manipon, Noel Amano, Céline Kerfant, Alfred Pawlik.

**Writing – review & editing:** Hermine Xhauflair, Noel Amano, Omar Choa, Alfred Pawlik.

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
