## [Decision Letter · Decision Letter 0]

26 Sep 2022

PONE-D-22-21195The invisible plant technology of Prehistoric Southeast Asia: Indirect evidence for basket and rope making at Tabon Cave, Philippines, 39-33,000 years ago.PLOS ONE

Dear Dr. Xhauflair,

Thank you for submitting your manuscript to PLOS ONE. After careful consideration, we feel that it has merit but does not fully meet PLOS ONE’s publication criteria as it currently stands. Therefore, we invite you to submit a revised version of the manuscript that addresses the points raised during the review process.

 This is an interesting paper and I look forward to seeing it in print.  However, I ask that you address all the points raised by the reviewers; in particulary review 2 who has provided detailed comments.  Please also add a section on the archaeological context of the material.  This is needed in order to demonstrate the justification for the early date.

We look forward to receiving your revised manuscript.

Kind regards,

Karen Hardy

Academic Editor

PLOS ONE

Journal Requirements:

2. In your manuscript, please provide additional information regarding the specimens used in your study. Ensure that you have reported specimen numbers and complete repository information, including museum name and geographic location.

For more information on PLOS ONE's requirements for paleontology and archaeology research, see https://journals.plos.org/plosone/s/submission-guidelines#loc-paleontology-and-archaeology-research.

Reviewers' comments:

Reviewer's Responses to Questions

**Comments to the Author**

1. Is the manuscript technically sound, and do the data support the conclusions?

Reviewer #1: Yes

Reviewer #2: Yes

2. Has the statistical analysis been performed appropriately and rigorously? 

Reviewer #1: N/A

Reviewer #2: I Don't Know

3. Have the authors made all data underlying the findings in their manuscript fully available?

Reviewer #1: Yes

Reviewer #2: No

4. Is the manuscript presented in an intelligible fashion and written in standard English?

Reviewer #1: Yes

Reviewer #2: Yes

5. Review Comments to the Author

Reviewer #1: I congratulate the Authors for this contribution on the perishable technology. The first Author has a long experience in experimenting soft materials and specifically of SE Asia. The MS raises still under investigated issues on plant's materials flexibility, pliability, twistability, resistance, all aspects that need testing before being dispatched as "cordage" or so.

I am pleased that the Authors took seriously these aspects and made an attempt to identify those features that may help in archaeological fibers characterization.

Some minor concerns I may raise are related to the whashing method devotrd to the archaeological residues (no mention in the MS IF residues resulted in the sonication solution).

In my experience I noticed that rattan has very characteristic phytoliths, so it can be easily recognized on stone tools.

line 445 ... of canna like dona and leaf I wonder if dona should be donax?

I would add a note to the graphic of the Figures: I would pay a little more attention in organizing the photos selecting a dimension that fits at best the single pic with the aim of making "order" within the single Figure. It will make the multiscale documentation ready appreciable.

Having said so, I am looking forward having this MS published in PlosOne.

Reviewer #2: To be accepted for publication in PLOS ONE, research articles must satisfy the following criteria:

1. The study presents the results of original research.

The results of use wear analysis on stone tools at Tabon Cave, Philippines, to identify processing tying/basketry material is the result of original research.

2. Results reported have not been published elsewhere.

So far as I am aware, the results have not been published elsewhere. As cited in the text, the results originate from PhD research, I encourage their publication.

3. Experiments, statistics, and other analyses are performed to a high technical standard and are described in sufficient detail.

Reference material: The results are based on the study of reference material in the form of stone tools used to process supple strips from ridged plant fibre, observations of tool use and reference material were obtained from experienced Pala’wan indigenous people (Ethics statement and in text). The effort taken to collect high quality reference collections of use wear by which to compare the archaeological use wear, is performed to one of the highest technical standards I have seen in use wear studies / traceology. The reference experiments are described in sufficient detail through text, images and supporting information.

Archaeology: Given that space is limited, it is understandable that the use wear images (Figures 2-6) are small. The benefit is that multiple views and wears are illustrated in each figure. The cleaning process and microscopy (p.8) is described. Proprietory software and coating materials should be named (l.168, 171).

Statistics: Table 1. Addition of standard deviation would help understand the range of tool size.

4. Conclusions are presented in an appropriate fashion and are supported by the data.

The discussion is sufficiently explanatory and broad ranging to address various points arising in the text. This leads into a summarising conclusion that is appropriate and provides a succinct overview of the results and their significance,

5. The article is presented in an intelligible fashion and is written in standard English.

The article is well written in standard English. In terms of sequence, it is awkward that the results for the experimental use wear analysis to be presented before the archaeological material. See p. 14. L.265, where is says (see below). However, I can also understand that to do this may involved lengthier description of the experimental traces. I suggest the authors consider sequence.

6. The research meets all applicable standards for the ethics of experimentation and research integrity.

The article achieves a high standard of the collection of data from indigenous people, as explained in the ethics statement and text.

7. The article adheres to appropriate reporting guidelines and community standards for data availability.

The inclusion of multiple images in the text will be useful for research critique and reproducibility. I have made suggestions on explaining coating and software above. Detailed additional data on the reference material collection and analysis is available in the supporting information.

The two Data Reviews are not accessible and is marked as ‘private’ when accessed through signed in Google account. The supplementary information in available.

Review

What are the main claims of the paper and how significant are they for the discipline?

This paper investigates the use wear on stone tools from Tabon Cave, Philippines, dated 39-33,000 BP. The wear is compared to a reference collection of use wear traces made by macro and microscopic examination of wear traces made by indigenous people of the same area working various materials. Through this analysis, the authors identify traces on three stone tools from Tabon Cave that compare with traces made when working rigid plant fibre materials to make supple strips of fibre. These results provide evidence of the role of organic materials, specifically plant fibres, in this early period in Southwest Asia. Early traces of fibre are very significance for the discipline, as these perishable materials are hard to trace archaeologically. They add important date on processing to a small body of evidence of plant fibre macro remains (as cited in the report) in the region.

The claim that ‘The paper provides new ways to identify organic technology in the archaeological record that is otherwise invisible’ in the abstract and p.2 line 37-38 needs to be reconsidered. The method of use wear analysis is well established. The novelty of this paper is its application to tropical plant taxa to create supple strips of fibre.

Are the claims properly placed in the context of the previous literature? Have the authors treated the literature fairly?

The claims are placed in the context of plant macro remains p.3. The literature is treated fairly. However, I recommend the reference to early flax in Dzudzuana Cave, Georgia is completely removed as this evidence has been seriously challenged (Bergfjord et al., 2010).

Bergfjord, C., Karg, S., Rast-Eicher, A., Nosch, M.-L., Mannering, U., Allaby, R. G., Murphy, B. M. & Holst, B. 2010. Comment on "30,000-year-old wild flax fibers". Science, 328, 163

Do the data and analyses fully support the claims? If not, what other evidence is required?

The data and analyses fully support the claims.

PLOS ONE encourages authors to publish detailed protocols and algorithms as supporting information online. Do any particular methods used in the manuscript warrant such treatment? If a protocol is already provided, for example for a randomized controlled trial, are there any important deviations from it? If so, have the authors explained adequately why the deviations occurred?

The supporting information is detailed and provides a substantive support for the method of data collection and wear analysis on stone tools.

I cannot see any supporting data on the archaeological context.

I cannot access URLs on youtube because they are private.

If the paper is considered unsuitable for publication in its present form, does the study itself show sufficient potential that the authors should be encouraged to resubmit a revised version?

NA

Are original data deposited in appropriate repositories and accession/version numbers provided for genes, proteins, mutants, diseases, etc.?

NA

Does the study conform to any relevant guidelines such as CONSORT, MIAME, QUORUM, STROBE, and the Fort Lauderdale agreement?

NA

Are details of the methodology sufficient to allow the experiments to be reproduced?

Yes, see supporting information.

Is any software created by the authors freely available?

NA. The proprietary software use should be named.

Is the manuscript well organized and written clearly enough to be accessible to non-specialists?

Yes. There are a few minor corrections:

L 62. String or sinew.

L 67.87. Consider ordering chronologically.

L 84-86. Be consistent throughout .[21] or [21].

L 93 Dye not die

L 121 karst network – explain

L 124 National Museum of where?

L 132 – Use name (not his) in first use in paragraph

L. 165. Why is reference [40] here? Do they have database standard. Be explicit.

L 173. Would be helpful to repeat date of site here.

L 218 Why is the goal to change the colour? What significance does colour change have?

L. 230 / 254 Table 1. It is important here to add tool type / morphology. Basic measurements – does this related to the terminology used in the supporting information? Description of basic measurements – does it mean tool measurements? Basic data. Basic to whom? Seems fundamental here. Stone tool data of Tabon Cave.

L 265 (see below) sequence to reconsider and avoid anticipating future text.

L 274 Add (Figure 4.g). I strongly recommend adding other in-text citations of figures parts.

L 297 add (Figure ???) as above

L 349 Add (figure ???) as above. Does this also need reference to supporting information.

L359 . Worked material .. which is? Name it here.

L 377. Can this be more closely related to the supporting information.

L387. Is resting percussion defined in the supporting information? If so, reference it here.

L396-8. This definition would benefit from being earlier.

L409 . Ring finger – suggest third finger.

L505 Does it need to be said that the tool is specifically for basketry and not other activities.

L595 Woven as past participle of weave, not weaved.

L608 Also hafting?

L613 Explain why the presence of phytoliths matters, e.g. because the silicious materials create wear.

L 635 Stone / lithic artefacts.

L638 After bamboo tools as the appropriate reference.

I recommend adding a short section on the archaeological context of the tools in Tabon cave. This is important especially to substantiate the early date.

Two papers which may be of interest (not essential to cite):

Gleba, M. & Harris, S. 2019. The first plant bast fibre technology: identifying splicing in archaeological textiles. Archaeological and Anthropological Sciences, 11, 2329–2346.

Hauser-Schäublin, B. 1996. The thrill of the Line, the String, and the Frond, or why the Abelam are a non-cloth culture. Oceania, 67, 81-106

Overall, in my opinion, this is a well-researched and documented experiment and analysis that comes to significant conclusions on the early use of plant fibres in Southwest Asia. The one significance addition I recommend is adding a short section on the archaeological context of the tools in Tabon cave. This is important to substantiate the early date. Otherwise, I recommend it for publication.

Is it your opinion that this manuscript contains an NIH-defined experiment of Dual Use concern?

No

Although confidential comments to the editors are respected, any remarks that might help to strengthen the paper should be directed to the authors themselves.

6. PLOS authors have the option to publish the peer review history of their article (what does this mean?). If published, this will include your full peer review and any attached files.

Reviewer #1: No

Reviewer #2: No

---

## [Author Response · Author response to Decision Letter 0]

18 Nov 2022

Response to reviewers:

We are very grateful to the editor and the two reviewers for their very constructive comments. Please find our responses below, in green. The changes are also made apparent in the manuscript entitled “with track change”. 

Reviewer #1: I congratulate the Authors for this contribution on the perishable technology. The first Author has a long experience in experimenting soft materials and specifically of SE Asia. The MS raises still under investigated issues on plant's materials flexibility, pliability, twistability, resistance, all aspects that need testing before being dispatched as "cordage" or so.

I am pleased that the Authors took seriously these aspects and made an attempt to identify those features that may help in archaeological fibers characterization.

Some minor concerns I may raise are related to the whashing method devotrd to the archaeological residues (no mention in the MS IF residues resulted in the sonication solution).

In my experience I noticed that rattan has very characteristic phytoliths, so it can be easily recognized on stone tools.

We did not collect the water resulting from sonication because the artefacts were from old excavations and we wanted to remove potential loose contamination. Nevertheless, we take note for the future that it could be a good thing to collect it and that rattan phytoliths are very recognisable. 

line 445 ... of canna like dona and leaf I wonder if dona should be donax?

Canna like dona is sometimes used as a common name to qualify Donax canniformis. For more clarity, we replaced it with the scientific name: Donax canniformis:

“We recorded the use of bamboo culms (erect and vine), stems of Donax canniformis and leaf petiole of palms to make weaving strips for basketry and traps.” 

Elsewhere in the text, canna like dona was replaced by Common Donax, a more frequent common name in English + the scientific name.

I would add a note to the graphic of the Figures: I would pay a little more attention in organizing the photos selecting a dimension that fits at best the single pic with the aim of making "order" within the single Figure. It will make the multiscale documentation ready appreciable.

The size of the pictures inside individual figures sometimes vary to best show relevant information or diagnostic features. If reviewer 1 would allow us to, we would like to keep the figures as they were originally designed. 

Having said so, I am looking forward having this MS published in PlosOne.

Reviewer #2: To be accepted for publication in PLOS ONE, research articles must satisfy the following criteria:

1. The study presents the results of original research.

The results of use wear analysis on stone tools at Tabon Cave, Philippines, to identify processing tying/basketry material is the result of original research.

2. Results reported have not been published elsewhere.

So far as I am aware, the results have not been published elsewhere. As cited in the text, the results originate from PhD research, I encourage their publication.

3. Experiments, statistics, and other analyses are performed to a high technical standard and are described in sufficient detail.

Reference material: The results are based on the study of reference material in the form of stone tools used to process supple strips from ridged plant fibre, observations of tool use and reference material were obtained from experienced Pala’wan indigenous people (Ethics statement and in text). The effort taken to collect high quality reference collections of use wear by which to compare the archaeological use wear, is performed to one of the highest technical standards I have seen in use wear studies / traceology. The reference experiments are described in sufficient detail through text, images and supporting information.

Archaeology: Given that space is limited, it is understandable that the use wear images (Figures 2-6) are small. The benefit is that multiple views and wears are illustrated in each figure. The cleaning process and microscopy (p.8) is described. Proprietory software and coating materials should be named (l.168, 171).

> Line 168: The software’s name (Leica Application Suite version 4.1.13.0) was added to the text. 

> In the method section as well, we added the following precisions about the coating of the artefacts: Line 171: “When residues were present, they were analysed in situ on uncoated stone tools, using a Field Emission Gun Scanning Electron Microscope (Quanta-650F) in high vacuum mode.”

Statistics: Table 1. Addition of standard deviation would help understand the range of tool size.

> We added standard deviation to Table 1

4. Conclusions are presented in an appropriate fashion and are supported by the data.

The discussion is sufficiently explanatory and broad ranging to address various points arising in the text. This leads into a summarising conclusion that is appropriate and provides a succinct overview of the results and their significance,

5. The article is presented in an intelligible fashion and is written in standard English.

The article is well written in standard English. In terms of sequence, it is awkward that the results for the experimental use wear analysis to be presented before the archaeological material. See p. 14. L.265, where is says (see below). However, I can also understand that to do this may involved lengthier description of the experimental traces. I suggest the authors consider sequence.

> The results of the archaeological material are actually presented first. Just below line 265 we only mention that similar use-wear was observed on experimental tools, whose description is further in the manuscript. 

 “These notches are associated with parallel striations perpendicular to the edge, as well as micro-polish located on the very edge and locally slightly invasive, expanding in perpendicular to the edge. This use-wear pattern was repeatedly observed on experimental stone tools used to thin plant fibres (see below).”

6. The research meets all applicable standards for the ethics of experimentation and research integrity.

The article achieves a high standard of the collection of data from indigenous people, as explained in the ethics statement and text.

7. The article adheres to appropriate reporting guidelines and community standards for data availability.

The inclusion of multiple images in the text will be useful for research critique and reproducibility. I have made suggestions on explaining coating and software above. Detailed additional data on the reference material collection and analysis is available in the supporting information.

The two Data Reviews are not accessible and is marked as ‘private’ when accessed through signed in Google account. The supplementary information in available.

That is very unfortunate. These files are two videos showing the gestures recorded among Pala’wan indigenous people and key moments of the experiments. 

We uploaded the videos on Youtube and are providing the URL in the paper. We hope that the files can also be uploaded on the Journal’s website, as an integral part of the paper. 

https://youtu.be/WLlUymCtSKQ

https://youtu.be/qiezpMVvQlI

Review

What are the main claims of the paper and how significant are they for the discipline?

This paper investigates the use wear on stone tools from Tabon Cave, Philippines, dated 39-33,000 BP. The wear is compared to a reference collection of use wear traces made by macro and microscopic examination of wear traces made by indigenous people of the same area working various materials. Through this analysis, the authors identify traces on three stone tools from Tabon Cave that compare with traces made when working rigid plant fibre materials to make supple strips of fibre. These results provide evidence of the role of organic materials, specifically plant fibres, in this early period in Southwest Asia. Early traces of fibre are very significance for the discipline, as these perishable materials are hard to trace archaeologically. They add important date on processing to a small body of evidence of plant fibre macro remains (as cited in the report) in the region.

The claim that ‘The paper provides new ways to identify organic technology in the archaeological record that is otherwise invisible’ in the abstract and p.2 line 37-38 needs to be reconsidered. The method of use wear analysis is well established. The novelty of this paper is its application to tropical plant taxa to create supple strips of fibre.

We replaced the text of lines 37-38 by : “This paper also provides a new way to identify supple strips of fibres made of tropical plants in the archaeological record, an organic technology that is otherwise most of the time invisible.”

Are the claims properly placed in the context of the previous literature? Have the authors treated the literature fairly?

The claims are placed in the context of plant macro remains p.3. The literature is treated fairly. However, I recommend the reference to early flax in Dzudzuana Cave, Georgia is completely removed as this evidence has been seriously challenged (Bergfjord et al., 2010).

Bergfjord, C., Karg, S., Rast-Eicher, A., Nosch, M.-L., Mannering, U., Allaby, R. G., Murphy, B. M. & Holst, B. 2010. Comment on "30,000-year-old wild flax fibers". Science, 328, 163

We are grateful to Reviewer 2 for this information. 

We read the response of Bergfjord and colleagues and as they conclude by : “Based on these two features, the fiber samples can be identified only as bast fibers. Any further conclusions about the nature of these fibers will require additional investigations with other techniques.”, we would like to replace the mention of flax in the manuscript by : 

“Likewise, 30 000-year-old bast fibres of an unknown species have also been found in Dzudzuana Cave, Georgia, associated with fungi and insects feeding on textiles[4,5]”

References 4 and 5 correspond to :

4. Kvavadze E, Bar-Yosef O, Belfer-Cohen A, Boaretto E, Jakeli N, Matskevich Z, et al. 30,000-Year-Old Wild Flax Fibers. Science. 2009;325: 1359–1359. doi:10.1126/science.1175404

5. Bergfjord C, Karg A, Rast-Eicher A, Nosch M-L, Mannering U, Allaby RG, et al. Comment on “30,000-Year-Old Wild Flax Fibers” | Science. Science. 2010;328: 1634. 

If bast fibres were found in a 30 000 year old site in Georgia, it is still interesting in our opinion, even if they do not correspond to flax. Nevertheless, we only know this discovery through literature. Does the revised text look acceptable to Reviewer 2? If not, we can remove the reference to the findings from Dzudzuana Cave.

Do the data and analyses fully support the claims? If not, what other evidence is required?

The data and analyses fully support the claims.

PLOS ONE encourages authors to publish detailed protocols and algorithms as supporting information online. Do any particular methods used in the manuscript warrant such treatment? If a protocol is already provided, for example for a randomized controlled trial, are there any important deviations from it? If so, have the authors explained adequately why the deviations occurred?

The supporting information is detailed and provides a substantive support for the method of data collection and wear analysis on stone tools.

I cannot see any supporting data on the archaeological context. 

I cannot access URLs on youtube because they are private.

> We uploaded the videos again on Youtube, made them public, and provided the link at different places of the paper. We really hope that they are now accessible. If not, please let us know and we will try a different method. 

> Archaeological context:

We added further information about the stratigraphic context and the new dates, including a new Figure showing a stratigraphic profile, the location of the charcoals dated and the location of the Assemblages II and III.

“More recently, it has been the subject of a reappraisal of the stratigraphy and Uranium series and new AMS 14C dating (Fig. 1).[26,30–32,34] This showed that the early stone tools known as Flake Assemblages II and III correspond to human use of the cave between 39 000- and 33 000-years BP, with an interruption sometime around 38 000 years BP due to increased hydrological activity and possible landscape changes.[26] The radiocarbon dates were obtained on five charcoal samples collected during the re-excavation of the cave in the 2000s and dated by Beta Analytic and Oxford. The detail of the reassessment of the chronological frame of the site will be the focus of another article. [35]”

Figure 1: Location of Assemblages II and III and the associated new 14C dates from charcoal (in pink) : 33.32 � 0.23 ka BP ( BETA-423462), 34.12 � 0.31 BP (BETA-261802), 35.42 � 0.35 ka BP ( BETA-259326), 37.60 �0.50 ka BP ( OxA-), 39.06 �0.37 ka BP (BETA-412819).These new dates push the age of the assemblages II and III further than previously thought based on older dates (in grey).[27,36] The detail of the new data on the chronological framework of Tabon Cave are the object of forthcoming publication. [26,35]

If the paper is considered unsuitable for publication in its present form, does the study itself show sufficient potential that the authors should be encouraged to resubmit a revised version?

NA

Are original data deposited in appropriate repositories and accession/version numbers provided for genes, proteins, mutants, diseases, etc.?

NA

Does the study conform to any relevant guidelines such as CONSORT, MIAME, QUORUM, STROBE, and the Fort Lauderdale agreement?

NA

Are details of the methodology sufficient to allow the experiments to be reproduced?

Yes, see supporting information.

Is any software created by the authors freely available?

NA. The proprietary software use should be named.

Is the manuscript well organized and written clearly enough to be accessible to non-specialists?

Yes. There are a few minor corrections:

L 62. String or sinew. 

We added the word “sinew”

L 67.87. Consider ordering chronologically.

We ordered the findings in the following order: 

1) Direct findings of fiber technology in Southeast Asia, ordered chronologically

2) Indirect findings in Southeast Asia, ordered chronologically as well. 

L 84-86. Be consistent throughout .[21] or [21].

We homogenised the position of the references after the point.

L 93 Dye not die > Corrected

L 121 karst network – explain > We removed “and the surrounding karst network complex” because it is in Tabon in particular that there is evidence for an old occupation was found.

L 124 National Museum of where? > Of the Philippines. Corrected

L 132 – Use name (not his) in first use in paragraph > Done 

L. 165. Why is reference [40] here? Do they have database standard. Be explicit.

The reference refers to the use-wear attribute recorded. We rephrased the sentence to make it more explicit: 

“The use-wear attributes recorded using a digital data base are exposed in details in the Supporting information p.8-13 and in a previous publication [41].”

L 173. Would be helpful to repeat date of site here.

We added the date:

“In this paper we present the results of the analysis of three artefacts from Tabon Cave, Flake Assemblages II and III, dated between 39 and 33 000 BP, which show diagnostic wear traces that correspond in distribution and morphology”

L 218 Why is the goal to change the colour? What significance does colour change have?

This is done for aesthetic reasons on some of the strips. 

We added the following text to the caption of Figure 1: “. The goal is to change the colour of the final product (strips used for weaving) for aesthetic purposes.”

L. 230 / 254 Table 1. It is important here to add tool type / morphology. Basic measurements – does this related to the terminology used in the supporting information? Description of basic measurements – does it mean tool measurements? Basic data. Basic to whom? Seems fundamental here. Stone tool data of Tabon Cave.

We removed the term “basic”. The measurements refer to the experimental stone tools: 

“Table 1: Mean and median values of measurements of the experimental tools. Details can be found in the Supporting information.”

L 265 (see below) sequence to reconsider and avoid anticipating future text.

Indeed there was a repetition here. We removed “This use-wear pattern was repeatedly observed on experimental stone tools used to thin plant fibres.” at the end of the paragraph as the information is already at the beginning. 

L 274 Add (Figure 4.g). I strongly recommend adding other in-text citations of figures parts.

L 297 add (Figure ???) as above

L 349 Add (figure ???) as above. Does this also need reference to supporting information.

We added in-text citations for the other figure parts as well.

L 349: We mention in the introduction to this section that the details can be found in the Supplementary information. We added more precisely which pages of the document:

“Further details of the use-wear observed on the 16 experimental tools used to perform this operation (as well as the terminology used) can be found in the Supporting information (p.14-46)”

L359 . Worked material .. which is? Name it here.

We rephrased the sentence : “The morphology of these striations varies in relation with the processed materials (bamboo, rattan, and Donax).”

L 377. Can this be more closely related to the supporting information.

> We specified the exact page numbers of the supplementary information related to this figure: “(see Supporting information p.14-46 for the details)”

L387. Is resting percussion defined in the supporting information? If so, reference it here.

> Yes. We added: “(see Supporting information p. 3-7 for a definition of the terms used)”

L396-8. This definition would benefit from being earlier.

> This is the beginning of the text devoted to the modern processing of plant fibres using metal tools. All the text before that refers to the activity being carried out with stone tools. The definition of the metal tools used by our Pala’wan informants can therefore not appear earlier in the text. 

L409 . Ring finger – suggest third finger. > It seems to us that 3rd finger is a bit confusing. We would prefer to keep “ring finger” for more clarity, although it is of course ethnically biased/connotated 

L505 Does it need to be said that the tool is specifically for basketry and not other activities. 

We are here referring specifically to the scrapping of the epidermis of 3 plant taxa. It is important to specify it because it is a different operation than the one described in most of the paper. 

L595 Woven as past participle of weave, not weaved.

Thank you very much, and sorry for that mistake. 

L608 Also hafting?

We did not do experiments of hafting but the topic has been very well covered by Veerle Rots who reported in details the resulting use-wear in several publications, for instance in : Rots, V., 2010. Prehension and hafting traces on flint tools : A methodology. Leuven University Press, Leuven.

L613 Explain why the presence of phytoliths matters, e.g. because the silicious materials create wear.

We added “very abrasive”: “This seems to be related to the fact that firstly, the gestures involved are different, and secondly, the physical properties of these plants are also different, particularly the hardness degree and the silica (very abrasive) and water contents.” The role of silica in polish formation could be the subject of several studies. 

L 635 Stone / lithic artefacts.

We replaced “stone artefacts” by “lithic artefacts”

L638 After bamboo tools as the appropriate reference.

We did not include references in the Conclusion section, as this is the usual practice.

I recommend adding a short section on the archaeological context of the tools in Tabon cave. This is important especially to substantiate the early date.

> We added more information on the new dates, as well as a stratigraphic section with the location of the charcoals dated and the assemblages. (see above and Lines 176-196 of the manuscript + the new Figure 1)

Two papers which may be of interest (not essential to cite):

Gleba, M. & Harris, S. 2019. The first plant bast fibre technology: identifying splicing in archaeological textiles. Archaeological and Anthropological Sciences, 11, 2329–2346.

Hauser-Schäublin, B. 1996. The thrill of the Line, the String, and the Frond, or why the Abelam are a non-cloth culture. Oceania, 67, 81-106

We sincerely thank Reviewer 2 for recommending these very interesting articles. 

Overall, in my opinion, this is a well-researched and documented experiment and analysis that comes to significant conclusions on the early use of plant fibres in Southwest Asia. The one significance addition I recommend is adding a short section on the archaeological context of the tools in Tabon cave. This is important to substantiate the early date. Otherwise, I recommend it for publiction.

> We added more information on the new dates, as well as a stratigraphic section with the location of the charcoal and the assemblages. (see above and Lines 176-196 of the manuscript + the new Figure 1)

Changes in the reference list

We added the following references:

-Based on a suggestion of Reviewer 2:

5. Bergfjord C, Karg A, Rast-Eicher A, Nosch M-L, Mannering U, Allaby RG, et al. Comment on “30,000-Year-Old Wild Flax Fibers” | Science. Science. 2010;328: 1634.

-Because we added a section and a Figure on the sedimentary/chronological context: 

35. Choa O, Xhauflair H, Dizon E, Ronquillo W, Arzarello M, Corny J, et al. The earliest colonisation of the Philippine archipelago by Late Pleistocene Homo sapiens in northeastern Sundaland: new radiocarbon ages for the earliest known human occupations from Tabon Cave (Palawan, Philippines). Forthcoming. 

36. Lewis HA, Johnson KR, Ronquillo W. Preliminary results of speleothem dating from Tabon Cave, Palawan, Philippines: moisture increase at the Last Glacial Maximum. Hukay. 2008.

These additions changed the numbering of almost all the references. 

Is it your opinion that this manuscript contains an NIH-defined experiment of Dual Use concern?

No

Although confidential comments to the editors are respected, any remarks that might help to strengthen the paper should be directed to the authors themselves.

6. PLOS authors have the option to publish the peer review history of their article (what does this mean?). If published, this will include your full peer review and any attached files.

Do you want your identity to be public for this peer review?For information about this choice, including consent withdrawal, please see our Privacy Policy.

Reviewer #1: No

Reviewer #2: No

---

## [Editor Report · Decision Letter 1]

6 Dec 2022

PONE-D-22-21195R1The invisible plant technology of Prehistoric Southeast Asia: Indirect evidence for basket and rope making at Tabon Cave, Philippines, 39-33,000 years ago.PLOS ONE

Dear Dr. Xhauflair,

Thank you for submitting your manuscript to PLOS ONE. After careful consideration, we feel that it has merit but does not fully meet PLOS ONE’s publication criteria as it currently stands. Therefore, we invite you to submit a revised version of the manuscript that addresses the points raised during the review process.

I am happy with the academic content of the ms.  But there needs to be a tightening up of the language throughout.  In some cases, the language is incorrect, in other cases, it is so convoluted that at times I had to read it several times, or refer back to other parts of the ms, to understand the point.  I have made a list of a few places that need corrections, but this is not exclusive. You need to go through the entire ms in detail and ensure that it is corrected.  I will not be able to accept it until this is done.

First page Introduction – not a good sentence.   Do you mean –Raw materials extracted from both plants and animals  constitute ….

Line 210 change ‘frame’ to ‘framework’

Line 227 - a 3 month fieldwork season, or 3 months of fieldwork

Be consistent in the use of capital letters in your Tables

Line 528: ‘Conclusion on the function of the analysed stone tools from Tabon Cave’  This is very confusing.  You should remove this heading, and simply move up the label Discusssion from below.  

Discussion.  We know you will be talking about your study so you do not need to state this.

You will probably need to reorder the first part of this. 

Line 645 ‘evidenced’ is incorrect. There is no verb ‘to evidence’. I suggest ‘indicated’

Line 682 It remains to explore and following…. Needs to be rewritten.

Also, what has watch making in Switzerland to do with this paper? This needs to be removed.

Line 698 - It had been hypothesised – by whom? Reference (even if it is a repeat)

Line 699 ‘and that the later ‘ this is unclear. Please rewrite.

What is the Bamboo Hypothesis? This is not specifically described in the ms-do you mean reference 66  If you are going to use it, you need to explain this more clearly and re-reference it here

Line 720 – ON other continents. Much longer time …. than what?  Rewrite. Do you mean a long time? Or longer than previously thought – in which case needs reference to the previous thought.

Line 713 ‘what was the’ – remove ‘what was’

Line 786 – correct the spelling of . Ibannez

We look forward to receiving your revised manuscript.

Kind regards,

Karen Hardy

Academic Editor

PLOS ONE
---

## [Author Response · Author response to Decision Letter 1]

27 Dec 2022

Dear Dr Hardy, 

Thank you very much for your comments which helped us to improve the style of the paper. The whole text was revised by a professional copy-editor and proof reader: Mr Robert Morgan. In addition, we addressed each of the specific points you raised.

Please find below our responses to your specific comments (in green in the Word version of this response).

Wishing you a Merry Christmas and a Happy New Year!

Hermine Xhauflair

First page Introduction – not a good sentence. Do you mean –Raw materials extracted from both plants and animals constitute ….

We changed into: “Plants constitute with animals the sources of raw materials of what Hurcombe calls “the missing majority”.”

Line 210 change ‘frame’ to ‘framework’ > Done

Line 227 - a 3 month fieldwork season, or 3 months of fieldwork

We changed into “a 3 month fieldwork season”

Be consistent in the use of capital letters in your Tables > Each entry is now starting awith a capital letter. 

Line 528: ‘Conclusion on the function of the analysed stone tools from Tabon Cave’ This is very confusing. You should remove this heading, and simply move up the label Discusssion from below. 

This is really part of the Results section:

1) The use-wear and residues observed on artefacts are presented, 

2) We expose comparable use-wear on experimental tools

3) We show the ethnographic observations based on which the experiments were designed.

4) We get back to the archaeology artefacts showing what we can infer about their utilisation based on the experimental and ethnographic data. 

We are offering to rename the last section :

“Understanding further the function of Tabon Cave artefacts based on experimental and ethnographic data”

Discussion. We know you will be talking about your study so you do not need to state this. You will probably need to reorder the first part of this. .

We changed the text into: “Diagnostic use-wear pattern characteristic of thinning plant fibres were determined on three stone tools from Tabon Cave dating back 39-33,000 years. We were able to identify this pattern because it is identical to the use-wear distribution observed”

And into: For many decades, researchers working in Southeast Asia have formulated the hypothesis that simple and non-standardised lithic artefacts had been complemented by plant implements made of bamboo.[44,67–71] In contrast, our results constitute evidence for the existence of perishable plant-based technology in prehistoric Palawan, even though the number of archaeological artefacts displaying this thinning pattern is still limited. They show that early plant technology did indeed exist in tropical regions during the Pleistocene, but it was not focused simply on the manufacturing of bamboo tools. This complements our discovery that some of the denticulates from Tabon Cave had been used to split plants, an operation that is part of the manufacturing of many objects nowadays, including flooring, musical instruments, and darts.

In fact, the human groups who inhabited Tabon Cave had developed a botanical knowledge deep enough to know which plants within their environment had fibrous, flexible, and solid properties, and could be turned into ropes, baskets, and other fibrecraft.

As early as 39-33,000 years ago, the Late Pleistocene inhabitants of Palawan possessed an elaborate organic technology and were processing plant fibres to make cordage, baskets, traps, or other composite objects.

Line 645 ‘evidenced’ is incorrect. There is no verb ‘to evidence’. I suggest ‘indicated’

Done

Line 682 It remains to explore and following…. Needs to be rewritten. 

We rewrote this section into:

“More than 30 000 years later, these botanical knowledge and technological know-how are still alive and allow many communities all over Southeast Asia to produce objects necessary to answer their everyday needs in a sustainable way. It remains to be explored whether these contemporary practices are the fruits of a continuous tradition directly rooted into Late Pleistocene.”

Also, what has watch making in Switzerland to do with this paper? This needs to be removed. > We removed the reference to the tradition of watchmaking in Switzerland.

Line 698 - It had been hypothesised – by whom? Reference (even if it is a repeat)

Line 699 ‘and that the later ‘ this is unclear. Please rewrite.

What is the Bamboo Hypothesis? This is not specifically described in the ms-do you mean reference 66 If you are going to use it, you need to explain this more clearly and re-reference it here

We rewrote the paragraph into: 

“These results show that seemingly simple Southeast Asian artefacts are hiding testimonies of a behavioural complexity invisible to the naked eye but that we can reveal through use-wear analysis. It had been hypothesized by proponents of the Bamboo Hypothesis that Southeast Asian stone tools had been used mainly to manufacture bamboo tools and that the focus of prehistoric craft makers on this giant grass (Bamboo is a Poaceae) would explain the technological simplicity of lithic artefacts. [91,92,101,102] Our results add to the recent evidence showing that a plant-based perishable technology indeed existed in the region during Prehistory, but also add nuance to the Bamboo Hypothesis sensu stricto, showing that people invested on plant materials in a much broader sense and did not use their stone tools exclusively to make bamboo knives, arrows and darts.” 

1. Solheim WG. The “New look” of Southeast Asian prehistory. The Journal of the Siam Society. 1972;60: 1–20. 

2. Pope GG. Bamboo and human evolution. Natural history (USA). 1989 [cited 11 Apr 2014]. Available: http://agris.fao.org/agris-search/search.do?recordID=US9010157

3. Forestier H. Des outils nés de la forêt. De l’importance du végétal en Asie du Sud-Est dans l’imagination et l’invention technique aux périodes préhistoriques. Acte du Séminaire-atelier Orléans, 15 et 16 octobre 1998. IRD Editions. Paris: Froment A. & Guffroy J.; 2003. pp. 315–337. 

4. Forestier H. La pierre et son ombre : épistémologie de la Préhistoire / Hubert Forestier (2020) - Société Préhistorique française. L’Harmattan. Paris; 2020. Available: http://www.prehistoire.org/offres/gestion/actus_515_40990-633/la-pierre-et-son-ombre-epistemologie-de-la-prehistoire-hubert-forestier-2020.html

Line 720 – ON other continents. Much longer time …. than what? Rewrite. Do you mean a long time? Or longer than previously thought – in which case needs reference to the previous thought.

Line 713 ‘what was the’ – remove ‘what was’

We rewrote the end of the Conclusion into:

Elsewhere in the world, a cord fragment made of twisted bark and dating to 46 ± 5 ka to 52 ± 2 ka was found at the Abris du Maras in France, and pierced shells from the Grotte des Pigeons, Morocco, and Blombos, South Africa constitute indirect evidence for the use of strings respectively 82 000 and 75 000 years ago.[9,10] Future analyses of lithic artefacts from Southeast Asia using the experimental reference we provided here can reveal if the antiquity of fibre technology was even greater than 39-33 000 BP in this region, what was the geographical extent of this craft, and if its contemporary practice is the result of an uninterrupted tradition. 

Line 786 – correct the spelling of . Ibannez

Corrected into: J. Ibañez

In addition, two references were added. We realised that we had forgotten to include them in the previous versions of the manuscript:

9. Bouzouggar A, Barton N, Vanhaeren M, d’Errico F, Collcutt S, Higham T, et al. 82,000-year-old shell beads from North Africa and implications for the origins of modern human behavior. Proc Natl Acad Sci. 2007;104: 9964–9969. doi:10.1073/pnas.0703877104

10. d’Errico F, Henshilwood C, Vanhaeren M, van Niekerk K. Nassarius kraussianus shell beads from Blombos Cave: evidence for symbolic behaviour in the Middle Stone Age. J Hum Evol. 2005;48: 3–24. doi:10.1016/j.jhevol.2004.09.002

---

## [Editor Report · Decision Letter 2]

3 Jan 2023

PONE-D-22-21195R2The invisible plant technology of Prehistoric Southeast Asia: Indirect evidence for basket and rope making at Tabon Cave, Philippines, 39-33,000 years ago.PLOS ONE

Dear Dr.  Xhauflair,

Thank you for submitting your manuscript to PLOS ONE. After careful consideration, we feel that it has merit but does not fully meet PLOS ONE’s publication criteria as it currently stands. Therefore, we invite you to submit a revised version of the manuscript that addresses the points raised during the review process.

There remain considerable challenges to understand parts of this ms.  Rather than send it back to you to ask again for corrections, I have made the suggestions myself.  I have only focused on places where the language needs to be corrected to ensure that the meaning is understood.  

LINE 68,69. Quafzeh,  CORRECT SPELLING

Line 156 and line 202. ‘frame’ should be ‘framework’

Line 319 ‘On this ventral face, the polish is flat, shiny and covering, cut by dark striations’

You need to change this sentence to explain what you mean by ‘covering’.  What was covered?  Do you mean for example, that it covered the entire surface? If so, you need to explain this.  Please clarify and correct.

*Lines 423 ff Occasionally, we witnessed 424 this operation performed with a large machete (tukäw) when we were in the forest and under 425 circumstances in which someone wanted to show usindicate how this operation was done on 426 the spot, pointing out that the appropriate tool was normally the smaller knife called paqis 427 (that is usually kept at home and not carried along like thein the manner of a machete).*

This long sentence is very difficult to follow.

Suggest change to

Occasionally, in the forest we witnessed this operation performed with a large machete (tukäw). This was done when an individual summarily wanted to show us how this operation was done. However, on each occasion, they pointed out to us that the appropriate tool would normally be the smaller knife called *paqis* 427 (that is usually kept at home and not carried along in the manner of a machete).

Line 462. ‘During our stay, weaving baskets was always performed by women.’

This should be ‘*During our stay, basket weaving was always performed by women*.’

Line 467 ‘Even if making weaving strips is a female activity for the Pala’wan communities we stayed 468 with, It was men who also knew how to process the plant segments to make weaving 469 stripsfor this purpose because (the involving the same chaîne opératoire is the same than for 470 as making ties.) As a demonstration of this, and itthe person was a man, (Linggit Rilla,) who 471 showed us how to process the bamboo vine Dinochloa sp. and Donax canniformis was a man: 472 Linggit Rilla.’

*HOW CAN YOU BE SURE THAT THIS IS BECAUSE IT IS THE SAME CHAINE OPERATIORE? YOU NEED TO DEMONSTRATE THIS, OR REMOVE IT.*

This whole part is confusing and potentially problematic. Do you mean you had one person, a man, showing you how to make the weaving strips, yet you ‘know’ that in fact it was always women? How do you know? Why did only one person show you this and why, in this case, was it a man and not a woman, since this was a woman’s task?

This is very difficult to follow. Suggest the following

‘Though making weaving strips is a female activity for the Pala’wan communities we stayed 468 with, men also knew how to process the plant segments for weaving strips since the same chaîne opératoire was used to make ties. For example, the person who showed us how to process the bamboo vine Dinochloa sp. and Donax canniformis was a man: 472 Linggit Rilla.’

Line 501. 3.4. Understanding further the function of Tabon Cave 502 artefacts based on experimental and ethnographic 503 data. SUGGEST AS FOLLOWS;

3.4. DEVELOPING AN UNDERSTANDING OF the function of Tabon Cave 502 artefacts based on experimental and ethnographic 503 data

Line 554 Mainly the contact face but sometimes also the non contact face exhibit REMOVE S HERE directional markers in perpendicular and diagonal orientation to the edge; THESE are often numerous and indicate a repeated transversal motion. These directional markers consist REMOVE S HERE of VARIABLE striations. THESE CAN BE EITHER long and parallel **and can form larger sets parallel between them**  **or long trails that materialise the contact with the processed plant strips;**
**undulations of fluted 560 polish; , stretched-out polish components forming lines; , and/or polish expanding inside the 561 tool surface. **THE MEANING OF THE PART IN** BOLD** ESCAPES ME. PLEASE ADD IN AN EXPLANATORY DIAGRAM. PLEASE REMEMBER THAT THIS IS A BROAD JOURNAL, THE PEOPLE READING YOUR ARTICLE MAY NOT BE SPECIALISTS IN USE WEAR.

LINE 584 from Tabon Cave had been used to split plants, an operation that is part of the 585 manufacturing of many objects nowadays, including flooring, musical instruments, and darts.

I assume here you mean that the split plants are part of the many objects.  This is currently unclear. Suggest as follows:

Tabon Cave had been used to split plants. Split plants form component parts NOTE I HAVE ADDED AN S HERE of many objects nowadays, including flooring, musical instruments, and darts.

Line 592 ‘Establishing experimental reference collections of use-wear resulting from realistic activities 593 based on observations of skilled craft makers in ethnographic contexts is a powerful means 594 of recovering evidence of perishable material culture’

Clunky and hard to follow. Suggest as follows:

Establishing experimental USE WEAR reference collections CREATED AS A RESULT OF realistic activities 593 based on observations of skilled craft makers in ethnographic contexts, OFFERS A powerful TOOL TO  recover evidence of perishable material culture, AS WE DEMONSTRATE HERE, AND AS ADVOCATED ELSEWHERE … ’

Line 627 Archaeological evidence for these fishing practices since the Late 628 Pleistocene has been found

Suggest as follows: Archaeological evidence for these fishing practices has been found from the Late 628 Pleistocene onwards

Line 630.  ‘from the forest’, If not the forest, then where? Please clarify.

Line 636 More generally, tying materials allowed the development of all a great many kinds of 637 composite technologies.

SUGGEST AS FOLLOWS:

More generally, tying materials ENABLED THE DVELOPMENT OF AN EXTENSIVE composite technology.

LINE 638. WHAT IS ADDITIVE TECHNOLOGY? PLEASE CLARIFY AND CORRECT.

LINE 645 what was reported here. CHANGE TO ‘those we report here’

LINE 655 s had in fact the great technological expertise and the 656 advanced skills necessary for thinning plant fibres [

HERE AS YOU HAVE WRITTEN IT, IT IMPLIES THAT  THEIR GREAT TECHNOLOGICAL EXPERTISE IS *ONLY *THEIR advanced skills necessary for thinning plant fibres [75] and possibly for assembling strips to 657 manufacture three dimensional objects.. I DO NOT THINK YOU ACTUALLY MEAN THIS, BUT YOU NEED TO EXPLAIN WHAT YOU MEAN.

LINE 674 into the Late Pleistocene – SHOULD BE IN NOT INTO

LINE 689 It had been hypothesized by 690 proponents of the Bamboo Hypothesis that Southeast Asian stone tools had been used 691 mainly to manufacture bamboo tools a

SUGGEST AS FOLLOWS:  OUR RESUTS SUPPORT The Bamboo Hypothesis that Southeast Asian stone tools had been used 691 mainly to manufacture bamboo tools a

LINE 703 FF. THIS IS A REPETITION YOU DO NOT NEED TO REPEAT IT HERE. elsewhere in the world, a cord fragment made of twisted bark and dating to 704 46 ± 5 ka to 52 ± 2 ka was found at the Abris du Maras in France, and pierced shells from the 705 Grotte des Pigeons, Morocco, and Blombos, South Africa constitute indirect evidence for the 706 use of strings respectively 82 000,000 and 75 000,000 years ago.[9,10]

LINE 709 we provided – remove the d. should be provide

We look forward to receiving your revised manuscript.

Kind regards,

Karen Hardy

Academic Editor

PLOS ONE
---

## [Author Response · Author response to Decision Letter 2]

20 Jan 2023

Dear Dr Hardy, 

Thank you very much for taking the time to look in details into our manuscript, for helping us to improve it and for giving us an opportunity to learn during the process. Please find our responses to your proposed modifications below, in green.

There remain considerable challenges to understand parts of this ms. Rather than send it back to you to ask again for corrections, I have made the suggestions myself. I have only focused on places where the language needs to be corrected to ensure that the meaning is understood. 

LINE 68,69. Quafzeh, CORRECT SPELLING> We changed into Qafzeh

Line 156 and line 202. ‘frame’ should be ‘framework’ >done

Line 319 ‘On this ventral face, the polish is flat, shiny and covering, cut by dark striations’

You need to change this sentence to explain what you mean by ‘covering’. What was covered? Do you mean for example, that it covered the entire surface? If so, you need to explain this. Please clarify and correct.

>

This is a description of the degree of linkage of the polish. The categories we use to describe the polishes are detailed in the Supporting information, so we also referred to it. We changed the sentence into the following:

“On this ventral face the polish is flat, shiny, with a high degree of linkage (covering [59]-see the Supporting information p.11) , cut by dark striations”

Lines 423 ff Occasionally, we witnessed 424 this operation performed with a large machete (tukäw) when we were in the forest and under 425 circumstances in which someone wanted to show usindicate how this operation was done on 426 the spot, pointing out that the appropriate tool was normally the smaller knife called paqis 427 (that is usually kept at home and not carried along like thein the manner of a machete).

This long sentence is very difficult to follow.

Suggest change to

Occasionally, in the forest we witnessed this operation performed with a large machete (tukäw). This was done when an individual summarily wanted to show us how this operation was done. However, on each occasion, they pointed out to us that the appropriate tool would normally be the smaller knife called paqis (that is usually kept at home and not carried along in the manner of a machete).

> Thank you very much! It is much clearer this way. We changed accordingly. 

Line 462. ‘During our stay, weaving baskets was always performed by women.’

This should be ‘During our stay, basket weaving was always performed by women.’

> Thank you very much. We changed accordingly.

Line 467 ‘Even if making weaving strips is a female activity for the Pala’wan communities we stayed 468 with, It was men who also knew how to process the plant segments to make weaving 469 stripsfor this purpose because (the involving the same chaîne opératoire is the same than for 470 as making ties.) As a demonstration of this, and itthe person was a man, (Linggit Rilla,) who 471 showed us how to process the bamboo vine Dinochloa sp. and Donax canniformis was a man: 472 Linggit Rilla.’

HOW CAN YOU BE SURE THAT THIS IS BECAUSE IT IS THE SAME CHAINE OPERATIORE? YOU NEED TO DEMONSTRATE THIS, OR REMOVE IT.

This whole part is confusing and potentially problematic. Do you mean you had one person, a man, showing you how to make the weaving strips, yet you ‘know’ that in fact it was always women? How do you know? Why did only one person show you this and why, in this case, was it a man and not a woman, since this was a woman’s task?

This is very difficult to follow. Suggest the following

‘Though making weaving strips is a female activity for the Pala’wan communities we stayed 468 with, men also knew how to process the plant segments for weaving strips since the same chaîne opératoire was used to make ties. For example, the person who showed us how to process the bamboo vine Dinochloa sp. and Donax canniformis was a man: 472 Linggit Rilla.’

> Thank you very much again. This is a good formulation. We replaced the sentence accordingly.

> We know that weaving baskets is a female activity because that is what people told us repeatedly and because this is what we witnessed during a 3-month fireldwork and subsequent visits. This information was also published by Nicole Revel, an ethnographer who studied Pala’wan culture for more than 45 years. We added a reference to one of her books which mentions that fact as well. 

Line 501. 3.4. Understanding further the function of Tabon Cave 502 artefacts based on experimental and ethnographic 503 data. SUGGEST AS FOLLOWS;

3.4. DEVELOPING AN UNDERSTANDING OF the function of Tabon Cave 502 artefacts based on experimental and ethnographic 503 data

> Ok. We changed accordingly.

Line 554 Mainly the contact face but sometimes also the non contact face exhibit REMOVE S HERE directional markers in perpendicular and diagonal orientation to the edge; THESE are often numerous and indicate a repeated transversal motion. These directional markers consist REMOVE S HERE of VARIABLE striations. THESE CAN BE EITHER long and parallel and can form larger sets parallel between them or long trails that materialise the contact with the processed plant strips; undulations of fluted 560 polish; , stretched-out polish components forming lines; , and/or polish expanding inside the 561 tool surface. THE MEANING OF THE PART IN BOLD ESCAPES ME. PLEASE ADD IN AN EXPLANATORY DIAGRAM. PLEASE REMEMBER THAT THIS IS A BROAD JOURNAL, THE PEOPLE READING YOUR ARTICLE MAY NOT BE SPECIALISTS IN USE WEAR.

>There is already a visual explaining these features: Figure 6. 

>We added a reference to Fig 6 at the end of the paragraph. 

“[…] polish components forming lines, and/or polish expanding inside the tool surface. (Fig 6)”

LINE 584 from Tabon Cave had been used to split plants, an operation that is part of the 585 manufacturing of many objects nowadays, including flooring, musical instruments, and darts.

I assume here you mean that the split plants are part of the many objects. This is currently unclear. Suggest as follows:

Tabon Cave had been used to split plants. Split plants form component parts NOTE I HAVE ADDED AN S HERE of many objects nowadays, including flooring, musical instruments, and darts.

>We modified accordingly

Line 592 ‘Establishing experimental reference collections of use-wear resulting from realistic activities 593 based on observations of skilled craft makers in ethnographic contexts is a powerful means 594 of recovering evidence of perishable material culture’

Clunky and hard to follow. Suggest as follows:

Establishing experimental USE WEAR reference collections CREATED AS A RESULT OF realistic activities 593 based on observations of skilled craft makers in ethnographic contexts, OFFERS A powerful TOOL TO recover evidence of perishable material culture, AS WE DEMONSTRATE HERE, AND AS ADVOCATED ELSEWHERE … ’

>We replaced accordingly.

Line 627 Archaeological evidence for these fishing practices since the Late 628 Pleistocene has been found

Suggest as follows: Archaeological evidence for these fishing practices has been found from the Late Pleistocene onwards

>We changed accordingly. 

Line 630. ‘from the forest’, If not the forest, then where? Please clarify.

>We changed into:

“Kerfant [19] highlighted the interesting aspect that most of the plants used in seafaring come from forests away from the coast, implying the mastery of both marine and inland environments.”

Line 636 More generally, tying materials allowed the development of all a great many kinds of 637 composite technologies.

SUGGEST AS FOLLOWS:

More generally, tying materials ENABLED THE DVELOPMENT OF AN EXTENSIVE composite technology.

>We changed into: 

” More generally, tying materials enabled the development of composite technologies.”

(We do not know yet how extensive it was)

LINE 638. WHAT IS ADDITIVE TECHNOLOGY? PLEASE CLARIFY AND CORRECT.

>This is a term used by André Leroi-Gourhan to qualify rope and basket technology (the idea is that matter is added and not removed). We changed into “fibre technology” for more clarity. 

LINE 645 what was reported here. CHANGE TO ‘those we report here’ > Done

LINE 655 s had in fact the great technological expertise and the 656 advanced skills necessary for thinning plant fibres [

HERE AS YOU HAVE WRITTEN IT, IT IMPLIES THAT THEIR GREAT TECHNOLOGICAL EXPERTISE IS ONLY THEIR advanced skills necessary for thinning plant fibres [75] and possibly for assembling strips to 657 manufacture three dimensional objects.. I DO NOT THINK YOU ACTUALLY MEAN THIS, BUT YOU NEED TO EXPLAIN WHAT YOU MEAN.

>We added some examples of other technological skills and innovation from the region : 

“In line with other studies showing for instance that the earliest representation of a scene is from Sulawesi, or that people had a mastery of navigation (e.g. [18,85,86,101,102]), our results contribute to rehabilitate Southeast Asian prehistoric heritage by showing that prehistoric groups had in fact great technological expertise and advanced skills, including the ones necessary for thinning plant fibres [75] and possibly for assembling strips to manufacture three dimensional objects.”

We hope that it is clearer (and more correct) this way.

We added one new reference: 

102. Aubert M, Lebe R, Oktaviana AA, Tang M, Burhan B, Hamrullah, et al. Earliest hunting scene in prehistoric art. Nature. 2019 [cited 13 Dec 2019]. doi:10.1038/s41586-019-1806-y

LINE 674 into the Late Pleistocene – SHOULD BE IN NOT INTO

> We modified accordingly

LINE 689 It had been hypothesized by 690 proponents of the Bamboo Hypothesis that Southeast Asian stone tools had been used 691 mainly to manufacture bamboo tools a

SUGGEST AS FOLLOWS: OUR RESUTS SUPPORT The Bamboo Hypothesis that Southeast Asian stone tools had been used 691 mainly to manufacture bamboo tools a

> We cannot say that because our results nuance the Bamboo Hypothesis sensu stricto. Here it is really important to keep our argumentation as it is. 

LINE 703 FF. THIS IS A REPETITION YOU DO NOT NEED TO REPEAT IT HERE. elsewhere in the world, a cord fragment made of twisted bark and dating to 704 46 ± 5 ka to 52 ± 2 ka was found at the Abris du Maras in France, and pierced shells from the 705 Grotte des Pigeons, Morocco, and Blombos, South Africa constitute indirect evidence for the 706 use of strings respectively 82 000,000 and 75 000,000 years ago.[9,10]

> We removed this sentence.

LINE 709 we provided – remove the d. should be provide

Done.

ADDED REFERENCE: 

102. Aubert M, Lebe R, Oktaviana AA, Tang M, Burhan B, Hamrullah, et al. Earliest hunting scene in prehistoric art. Nature. 2019 [cited 13 Dec 2019]. doi:10.1038/s41586-019-1806-y

---

## [Editor Report · Decision Letter 3]

24 Jan 2023

The invisible plant technology of Prehistoric Southeast Asia: Indirect evidence for basket and rope making at Tabon Cave, Philippines, 39-33,000 years ago.

PONE-D-22-21195R3

Dear Dr. Xhauflair,

We’re pleased to inform you that your manuscript has been judged scientifically suitable for publication and will be formally accepted for publication once it meets all outstanding technical requirements.

Kind regards,

Karen Hardy

Academic Editor

PLOS ONE
---

## [Editor Report · Acceptance letter]

30 Mar 2023

PONE-D-22-21195R3 

The invisible plant technology of Prehistoric Southeast Asia: Indirect evidence for basket and rope making at Tabon Cave, Philippines, 39-33,000 years ago. 

Dear Dr. Xhauflair:

I'm pleased to inform you that your manuscript has been deemed suitable for publication in PLOS ONE. Congratulations! Your manuscript is now with our production department. 

Kind regards, 

on behalf of

Dr. Karen Hardy 

Academic Editor

PLOS ONE